# The longest-lasting 2023 western North American heat wave was fueled by the record-warm Atlantic Ocean

Hosmay Lopez [1] ✉, Sang-Ki Lee [1], Robert West [2], Dongmin Kim [2] & Liwei Jia[3]

According to the World Meteorological Organization, 2023 was ranked as the second warmest year in the global surface temperature record since 1850, setting warm surface temperature records over more than 20% of the global land surface. In particular, the southwestern United States (US) and Northern Mexico experienced their longest stretch of record-breaking heat wave, affecting over 100 million people, causing over 200 deaths, and $14.5 billion in economic loss. Here we show that the 2023 heat wave event was linked to a strong anticyclonic blocking pattern that persisted for more than six weeks across the western US. Regression analysis and atmospheric model simulations suggest that the anticyclonic pattern was ultimately forced by the extremely warm sea surface temperature in the Atlantic. The combination of a warm Atlantic and a developing Pacific El Niño significantly amplified regional heat waves, doubling their number, tripling their days, and increasing their duration by about 50%.

The year 2023 was the second warmest on records[1,2], only surpassed by 2024 (Monthly Global Climate Report for Annual 2024). Of particular notoriety was the boreal summer months (i.e., June-August), which were characterized as the hottest summer with more than 20% of the land surface setting extreme warm records[2] and multiple heatwave events over most continents[3], with several concurrent events simultaneously affecting multiple regions (Lembo et al.[4]). The year 2023 also produced the highest number of heat-related deaths in the United States (U.S.) in the 21st century, with 2325 deaths in association with severe heat waves[5]. One of these severe heat wave events occurred over the southwestern U.S. and Mexico, which extended from mid-June to early August, affected over 100 million people, and was responsible for 303 death that occurred in a span of just two weeks in Maricopa County, Arizona (https://www.maricopa.gov/1858/Heat-Surveillance). The compounded effect of extreme heat and drought was responsible for $14.5 billion in economic loss (https://www.ncei.noaa.gov/news/national-climate-202312), making this event the costliest weather and climate disaster of 2023 in North America. This event

featured prolonged extreme surface temperatures, with Phoenix, Arizona experiencing both the longest continuous stretch (31 days, from 30 June to 30 July) of daily maximum temperature exceeding 43.3 °C (110 °F) and the warmest nighttime minimum temperature on record (36.1 °C).

Excessive heat puts significant stress on human health, resulting in increased morbidity and mortality[6,7], with some of the most notorious events being responsible for hundreds and even thousands of deaths in the most extreme cases; for example, the 1980 US heat wave[8,9], the 1995 event in Chicago, Illinois[10], the 2003 European heat wave[11], the 2010 Russian event[12,13], the 2011 event over the Great Plains of the US[14–16], the 2021 northwestern North American event[17–22], the 2023 South American heat wave[2]. In fact, extreme heat is the leading cause of weather-related mortality in the US, topping other more notorious weather hazards, like tornadoes and hurricanes (US, https://www.weather.gov/hazstat/). Heat waves are also the leading cause of natural hazards-related deaths in Australia, accounting for more than 55% of the reported fatalities[23]. Moreover, these extreme heat wave events

[1]Atlantic Oceanographic and Meteorological Laboratory, NOAA, Miami, FL, USA. [2]Cooperative Institute for Marine and Atmospheric Studies, University of Miami, Miami, FL, USA. [3]Geophysical Fluid Dynamics Laboratory, National Oceanic and Atmospheric Administration, Princeton, NJ, USA. ✉ e-mail: Hosmay.Lopez@noaa.gov

have been observed more frequently in many regions[24], with a significant increase in the number and severity of heat waves in recent decades as a result of rising surface temperatures[25]. However, the effects of increasing temperatures on heat extremes go beyond changes in the mean climate and include shifts in the extremes as well[26,27], where the duration and frequency are expected to increase this century[28,29]. All of these effects are further exacerbated by a projected increased exposure to heat extremes due to population growth[30], increased urbanization, agricultural loss, and aridification[31–33].

In addition to the longer-term trends, understanding shorter term weather and climate variability is essential for improving heat wave predictions and future projections. Heat waves are often linked to large-amplitude atmospheric circulation patterns driven by quasi-stationary and propagating Rossby waves and their interaction with the overall synoptic flow, topography, and land-sea contrast[4]. These interactions result in persistent anticyclonic flow and blocking events[34], leading to flow stagnation and prolonged periods of clear sky, enhanced incoming solar radiation, drought conditions, and reduced soil moisture, all of which further exacerbates surface warming[35–37]. While these blocking patterns are part of the atmospheric synoptic circulation, significant effort has been undertaken to further understand longer-term drivers of these patterns, with the aim to improve their predictions beyond the weather forecast range. Slower-acting coupled atmospheric-land-ocean processes are often attributed to heat waves. For example, sub-seasonal variations in midlatitude atmospheric circulation patterns have been shown to precede heat waves over the US by 15–20 days[38]. Enhanced convective activity from the East Asian Monsoon was found to force a mid-latitude wave train across the Pacific, leading to enhanced blocking pattern, which promotes the occurrence of US Great Plains heat waves[15]. Others have shown that persistent midlatitude circulation patterns forced by tropical sea surface temperature (SST) anomalies are modulators for drought and extreme heat over the western US[39–41]. A recent work found that boreal summer tropical Atlantic SST anomalies modulate heat wave occurrences over North America[16]. In that work, it was found that a warmer tropical Atlantic enhances atmospheric convection over the Caribbean Sea and produces a Gill-type atmospheric response[42]. This, in turn, produces an anticyclonic Rossby wave source over the Great Plains, thus enhancing subsidence and significant surface warming, leading to heat domes. The aforementioned works, and many others, have provided a better understanding of the coupled climate system as it pertains to heat extremes. This collectively suggests that the inherently longer timescales of oceanic and land process variabilities could aid in extending the prediction of high-impact extreme events beyond the weather timescales. Although current coupled models tend to underestimate regional terrestrial temperature variability, decreasing prediction skill at longer lead-times[43].

Besides the multiple land temperature records that were set in 2023, global oceans also experienced record warm SST[44,45]. Of special notoriety were the record SSTs over the tropical Atlantic, which were up to one degree Celsius warmer than climatology. These warm oceanic temperatures were not just confined to the surface, as oceanic heat content was at record levels in 2023[44]. The North Atlantic has experienced a warm period since around 1995, owing to its relation to the positive phase of the Atlantic Multidecadal Variability[46,47]. In addition, an increase in the energy imbalance from an upsurge in greenhouse gasses[44,48,49] has contributed to a steady rise in ocean surface temperatures and heat content, among other effects[50–52]. Furthermore, the 2020 emission regulations from the International Maritime Organization aimed at reducing ship sulfate aerosol emissions may have contributed to the recent warm surface temperatures over the North Atlantic[53,54]. Warm North Atlantic SSTs have been shown to modulate local and remote atmospheric circulation, with significant influences on precipitation[47,55,56], tropical cyclone activity[57], as well as

extreme land surface temperatures[49,58], and SSTs over the Pacific Ocean[59]. The tropical Pacific was also characterized by warm SSTs associated with a growing El Niño[60] (i.e., the positive phase of the El Niño Southern Oscillation, ENSO). ENSO is one of the dominant modes of interannual climate variability, which has been shown to affect extreme surface temperatures through modulation of atmospheric circulation[61–63], including the occurrence of heat waves over North America[64,65].

This work suggests a physical link between the co-occurrence of the long-duration (i.e., several weeks) extreme heat over the southwest US and Mexico and the record warm North Atlantic SSTs and a growing El Niño in the Pacific. Thus, we hypothesize a physical connection that the extremely warm 2023 interbasin Pacific-Atlantic SSTs were responsible for the persistence of the longest-lasting heatwave in the region. For this, we use observational records and general circulation model experiments to show that the growth and persistence of this heat wave event were supported by remote forcing from the record warm SSTs in the Atlantic, a growing El Niño event in the Pacific, and the interbasin synergy effect of Pacific-Atlantic forcing. This interbasin synergy is in reference to the constructive interaction between the warm Atlantic and ENSO in modulating the heat wave occurrence and not to the active debate about tropical Pacific/Atlantic interaction conundrum[66,67].

## Results
### Observed evolution of the heatwave
The maximum and minimum near-surface temperatures along with their climatological mean and 95th percentiles are shown in Fig. 1a for the summer of 2023 for Phoenix, Arizona (see Methods for definition of climatology). Note that the conditions were near their climatological mean for most of June. However, starting around 1 July, both maximum and minimum temperatures were at or exceeded their 95th percentile threshold (T95) for several days. The brunt of the heatwave was experienced from 13 July to 30 July, with 18 consecutive days of maximum temperatures exceeding the T95 by as much as 4.5 °C. Beyond that, four other shorter-duration extreme heat periods were experienced up to 10 September, where the temperatures reverted to their climatological mean. Besides being the longest-lasting heatwave event in the region, the 13–30 July 2023 event was also responsible for the warmest minimum temperature and tied for the third warmest maximum temperature on record (Fig. 1b, maximum temperatures of 50 °C and 49.4 °C were recorder in 1990 and 1995). While the southwest US and Mexico are notorious for extreme, persistent heat, previous events were less severe, with the previous record for Phoenix of consecutive days above T95 being 8 days, less than half the duration of the 2023 event. During the peak of the event, warm temperature anomalies >5 °C affected most of the region (Fig. 1c). Similar conditions were observed in other metropolitan areas, such as Las Vegas, Nevada, Albuquerque, New Mexico, El Paso, Texas, and San Antonio, Texas (Supplementary Fig. 1).

The spatiotemporal evolution of the maximum surface temperature anomaly during the heatwave is shown in Supplementary Fig. 2. Note that significant positive temperature anomalies were already present over most of Mexico as early as mid-June, coinciding with negative anomalies over the western US. The warm anomalies subsided by early July, but then reappeared around the second week of July, engulfing most of the western US in the second half of the month. The event then shifted back south in early August and finally progressed eastward over the Great Plains by mid-August, where several record temperatures were broken in other regions (e.g., New Orleans experienced a maximum temperature of 40.5 °C on 27 August).

### Heat Budget
The physical processes responsible for the growth and long persistence of the temperature anomalies over the southwest US are

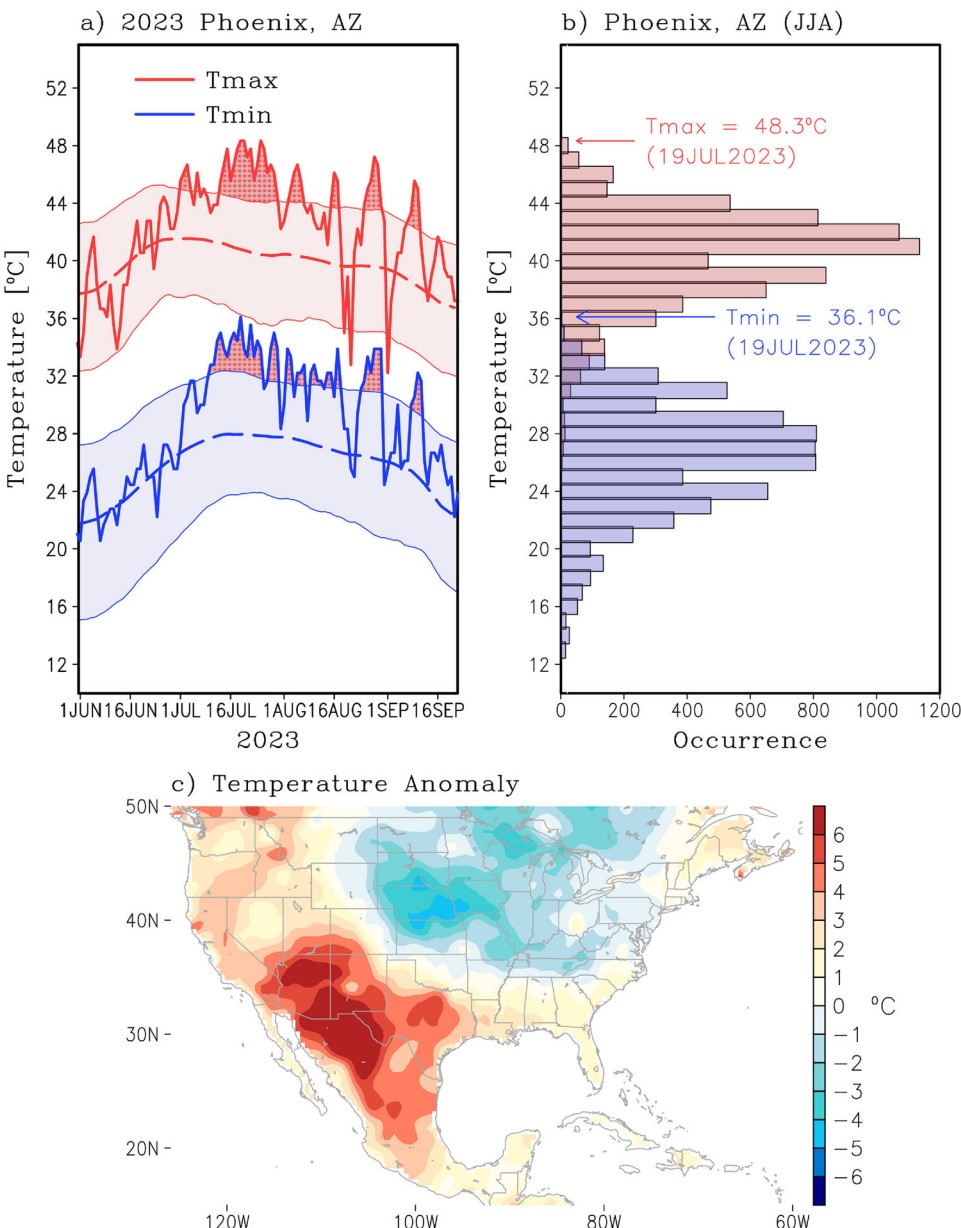

**Fig. 1 | Heat wave event of 2023. a** Seasonal evolution of maximum (red) and minimum (blue) temperature for Phoenix Arizona for the year 2023 from 1 June to 20 September. The long-term daily mean is shown by the dashed line, whereas the 5th and 95th percentiles are shown by the shading region. Excess above the 95th percentile is shown by red shading and stipples for both maximum and minimum temperatures. **b** Observed histogram of maximum (red) and minimum (blue) temperature for Phoenix, Arizona for the period 1 June to 31 August from 1955 to 2023. The extremely warm temperatures of 19 July 2023, during the peak amplitude of the heat wave, are shown for reference. **c** Five-day averaged surface temperature anomaly centered on 19 July 2023. Source data are provided as a Source Data file.

investigated next. A key question here is, why did the extreme heat last over twice as long as other previous events? Fig. 2 describes the temporal evolution, averaged every 10 days for easier illustration, of several relevant dynamical and thermodynamic variables from June-August and averaged from 30°–35°N to 115°–110°W, representing a box approximately encompassing the state of Arizona. A negative 200 hPa geopotential height anomaly was present during early June along with a negative 850 hPa temperature and 2-m temperature anomalies, which rapidly degraded and turned positive. These anomalies reached their maximum around mid-July, coinciding with the peak of the heatwave, and remained positive for the rest of the summer. The energy budget (see Methods and Eq. 4) can be used to discern the driving process that led to the rapid growth of the positive low-level temperature anomaly. For example, for most of June, there were significant positive net shortwave, longwave radiations, and

sensible heat fluxes at the surface (values > 20 Wm²). This coincided with enhanced vertically integrated (975–800 hPa, see Methods) heating rates (heating > 2 °C day⁻¹) due to vertical advection and to enhanced surface heating. By early July, most of the strong heating anomalies weakened, but remained slightly positive. However, significant surface sensible heat flux (>20 Wm²) and vertical diffusive heating rates (heating > 2 °C day⁻¹) supported the continuation of the $T_{850}$ anomalies throughout July, along with longwave radiation heating during August. Of note is that surface latent heat flux was negative throughout the summer, enhancing the surface warming through radiative and sensible heating via an increase in the Bowen ratio, which links water and energy balances of the climate system. A Bowen ratio increase, often present during mega heat waves[68], suggests a reduction in the evaporation/evapotranspiration and an increase in the sensible heating and thus increase in near surface temperatures.

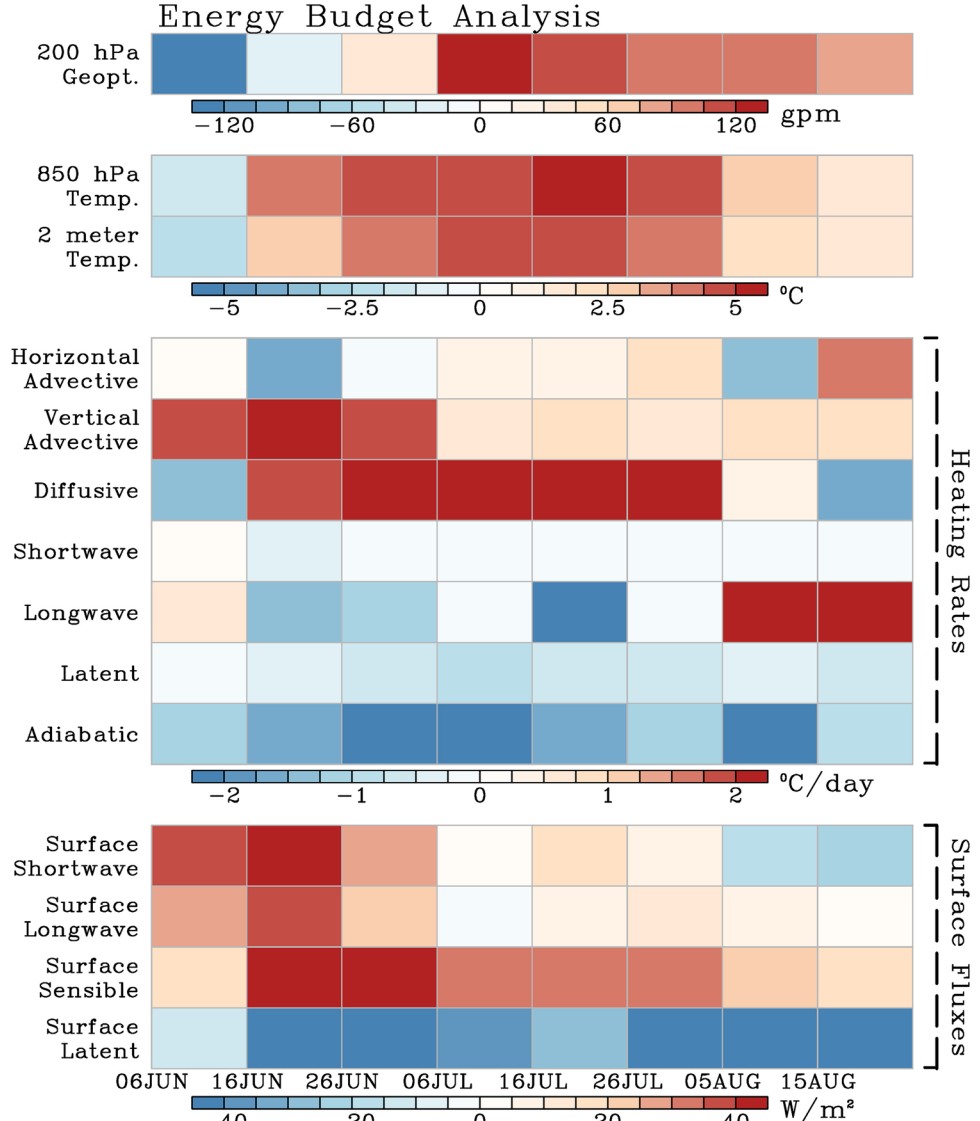

**Fig. 2 | Heat budget analysis.** Energy budget averaged every 10 days over the southwest U.S. and northwest Mexico (30°–35°N and 115°–110°W) from 6 June to 24 August 2023. Each row represents (from the top): 200 hPa geopotential height anomaly [gpm], 850 hPa temperature and 2-m temperature anomalies [°C], vertically integrated anomalous heating rates from 975 to 800 hPa [°C day⁻¹], and surface heat fluxes (net surface shortwave and longwave radiation, sensible and latent heat fluxes) [Wm⁻²]. Daily anomalies are derived from the long-term monthly mean for the 1979–2022 period. See Methods for heat budget definitions.

In contrast, adiabatic heating anomaly was fairly negative throughout the summer, lessening the impact of the remaining heating terms and thus dampening the temperature tendency. In addition, heating due to horizontal advection played a relatively small role in the evolution of the temperature anomaly, this is expected due to the broad temperature anomaly (Fig. 1c) and stagnant flow pattern (Fig. 3) associated with this heat wave. Overall, the low-level temperatures warmed significantly during June to mid-July, and then remained elevated for most of the summer, aided by significant surface heat fluxes and vertical heating rates (i.e., vertical advective, diffusive, and longwave heating).

We now turn our attention to the spatial distribution of the circulation anomalies during the peak of the heatwave. The analysis uses potential vorticity (PV) and circulation on constant potential temperature (i.e., isentropic surface), which poses several advantages over pressure surfaces in that PV can be used as a parcel tracer and is conserved for frictionless adiabatic motion[69]. Figure 3a depicts the 18 July 2023 potential vorticity and wind at the 350 K isentropic level, as well as the location of the dynamical tropopause (thick black contour),

defined to be the 2 PVU (2·10⁻⁶ K m² kg⁻¹ s⁻¹) iso-surface, which serves as the boundary of tropospheric and stratospheric air. Note that the flow is anticyclonic over the southwestern U.S and northern Mexico, with significant fluid trapping (thick arrows), suggesting a blocking pattern (see Methods for definition on trapped flow). In fact, the upstream and downstream troughs, along with the anticyclone, are manifestations of a classic "omega" blocking pattern, which is well known to persist for significant periods of time and is responsible for stagnant flow patterns. Regions under anticyclonic trapping often experience significant surface shortwave heating due to clear skies (Supplementary Fig. 3). These conditions contribute to extreme surface warming for prolonged periods, with little ventilation due the trapped air mass and flow stagnation. While atmospheric blocking is a manifestation of planetary waves[70], the aforementioned anticyclonic trapping persisted for several weeks, as shown by a time-averaged flow (Supplementary Fig. 4).

A latitude-vertical cross-section along 112°W, which is approximately through the center of the anticyclone, shows the anomalous circulation features during the peak of the heatwave (Fig. 3b). Note

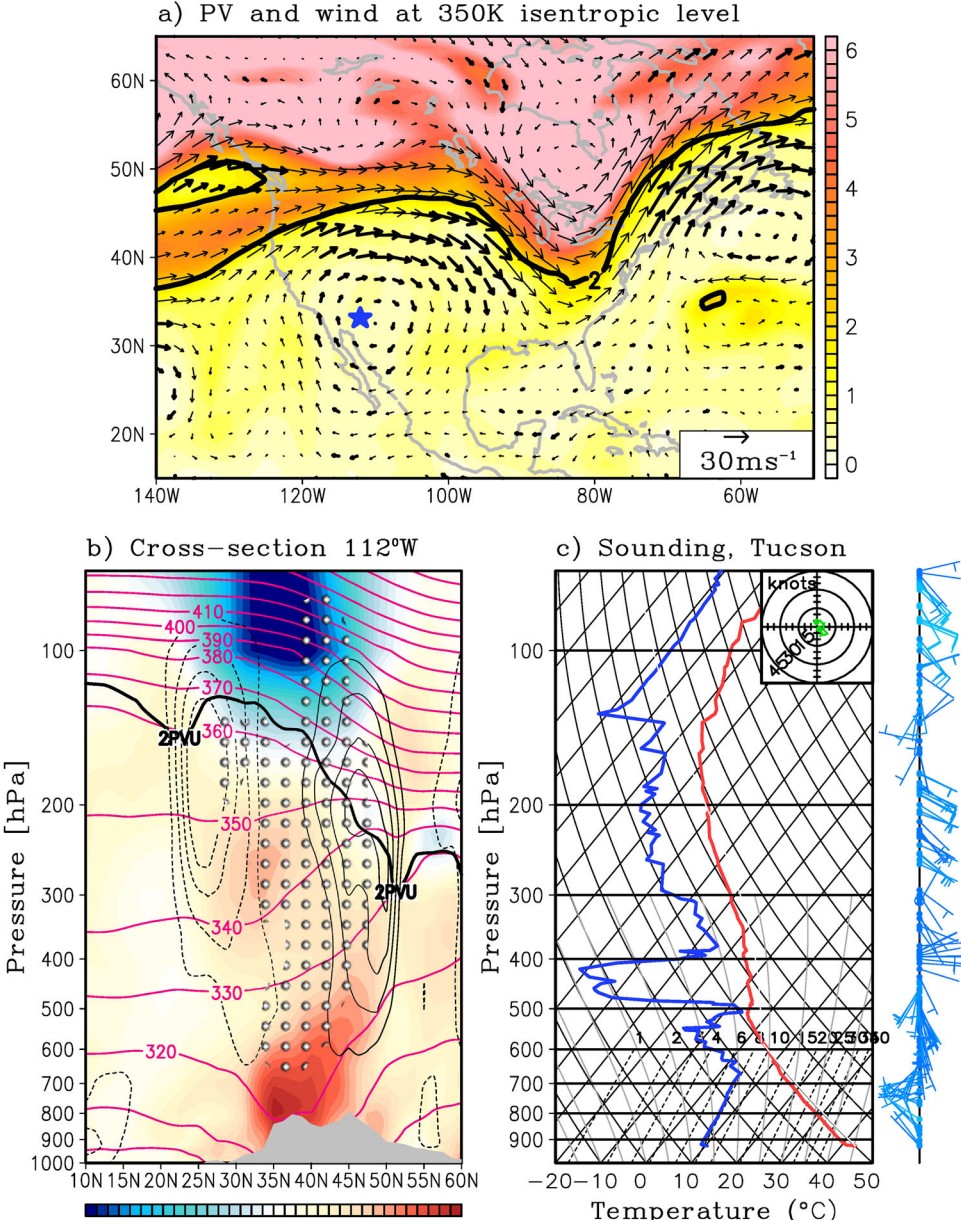

**Fig. 3 | Synoptic circulation during the peak of the heat wave. a** Potential vorticity and wind at the 350 K isentropic level during the maximum amplitude of the heat wave on 18 July 2023. Thick vectors depict anti-cyclonic fluid trapping, a proxy for heat dome and air flow stagnation. The thick black line indicates the location of the dynamical tropopause. The blue star on each panel represents the location of Phoenix, Arizona. **b** Latitude-vertical cross-section along 112°W on 18 July 2023 of anomalous potential temperature (color) and potential temperature (magenta 10 K intervals). Also shown are zonal wind anomalies (light black contours at 3 m/s intervals), the tropopause level as measured by the 2 PVU (thick black line), and anti-cyclonic fluid trapping (circle hatching at $10^{-5}\,s^{-1}$). **c** Vertical atmospheric profile over Tucson, Arizona. The profile is plotted on a skewT-logP thermodynamic diagram. The vertical axis is pressure (hPa), the skewed thin axis is temperature (Celsius), and the dry and moist adiabats are also shown. The vertical profile of environmental temperature, dew point, and wind speed and direction is denoted by the thick-red line, thick-blue line, and wind barbs, respectively.

that there is a deep upper-level (700–100 hPa) anticyclonic circulation centered around 35°N with easterlies (westerlies) around 25°N (45°N). The 2 PVU contour shows a dome-like feature, with PV decreasing northward of 25°N (i.e., a meridional PV inversion) with a subsequent poleward increase. In addition, there is a significant downward (upward) bulging of the potential temperature surfaces at lower (upper) levels, indicating low to mid-level heating. This is shown by the potential temperature anomalies (colors in Fig. 3b), indicating potential temperature anomalies around 8 °C near the surface and similar amplitude negative anomalies in the stratosphere. In relation to the strong mid-level anticyclone, the air mass from 25°N to 50°N is

significantly trapped (hatching in Fig. 3b, see Eq. 5 for definition) for most of the tropospheric depth, providing little to no ventilation and thus exacerbating the lower-level warm anomalies. In the presence of a trapped flow, most of the ventilation comes from vertical advection from the boundary layer up, which is the case here as shown in the heat budget analysis (Fig. 2). The vertical profile over Tucson, Arizona on 18 July 2023 (Fig. 3c) shows a well-mixed lower troposphere, with a deep dry layer of constant potential temperature (i.e., red line and potential temperature lines are parallel) and mixing ratios (i.e., blue line and dashed black lines are parallel) from the surface up to about 650 hPa, indicating strong vertical turbulent heat and moisture fluxes

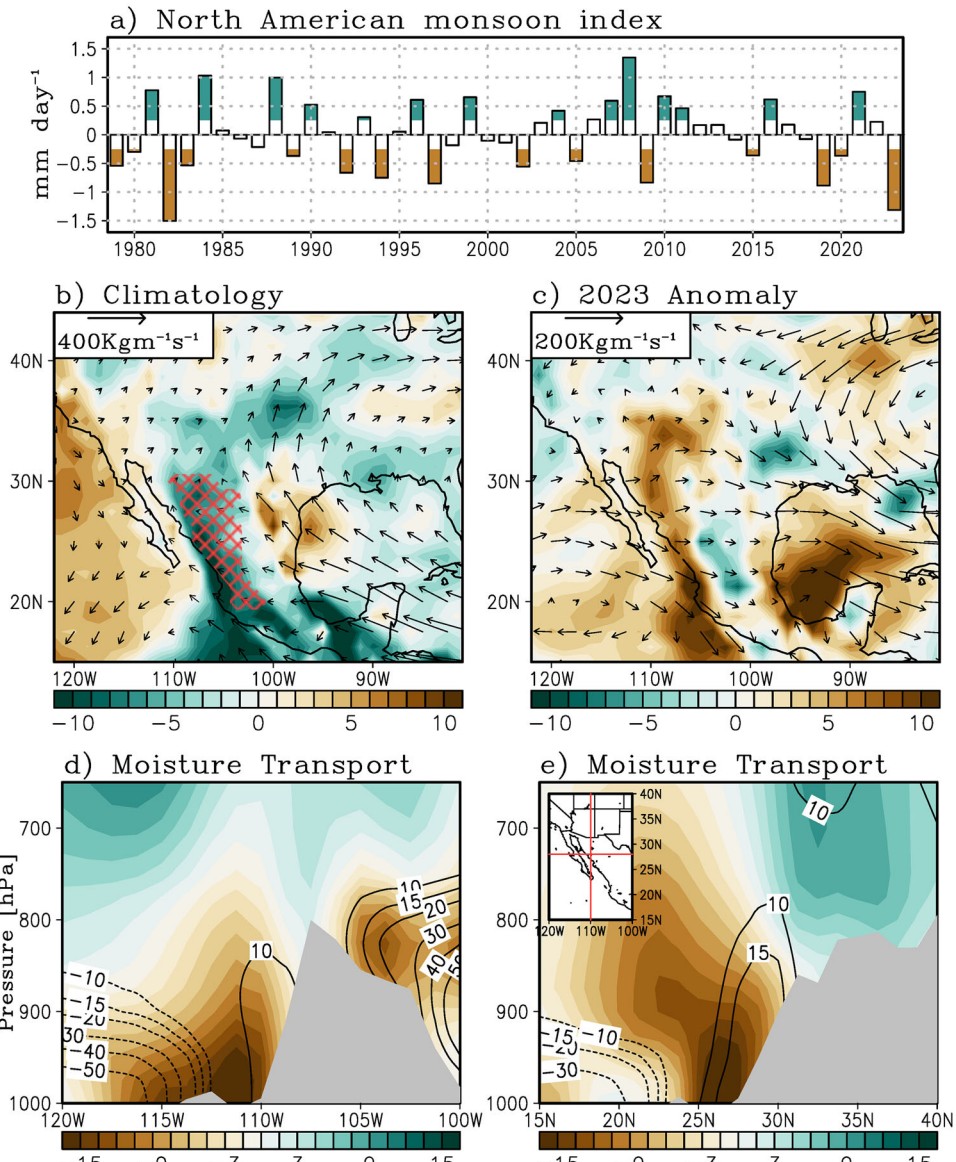

**Fig. 4 | The North American Monsoon. a** Time series of the North American monsoon (NAM) index for June–August. Years of significant NAM index anomalies are highlighted by color-filled bars based on the 99-percentile significance level based on the student-T test. **b** Climatological vertically integrated moisture transport (vector, Kg m⁻¹ s⁻¹) and its divergence (color, Kg m⁻² s⁻¹), where negative values indicate convergence. **c** Same as (**b**) but for the 2023 anomalies computed from the departure from the 1979–2022 climatology. **d** Zonal-vertical cross-section along 28°N of the moisture transport climatology (black contours, solid lines indicate positive values, dotted lines indicate negative values) and 2023 anomaly (color). **e** Same as **d** but for the meridional cross-section along 110°W. Red hatching in **b** indicates the region that meets the criteria for the NAM monsoon index (see Methods).

associated with enhanced surface heating from prolonged clear sky conditions. The profile also shows a marked decrease in dew point temperature above 500 hPa, indicative of subsidence associated with the high-pressure dome as well as weak wind speed and weak vertical wind shear (hodograph in the top right of Fig. 3c).

**A weak North American monsoon in 2023**

The southwest U.S. and northern Mexico are regions where most of the precipitation and thus soil moisture is obtained during the summer monsoon[71–73] (i.e., North American monsoon, NAM, see Methods for definition). In addition, soil moisture and surface air temperature are strongly correlated through longwave radiation, sensible, and latent heating through land-atmospheric coupling[74], thus a moisture deficit could exacerbate warm surface temperatures and heat wave occurrence[36,75]. Thus, it is important to assess the state of the NAM,

which is the dominant source of precipitation over the region. Note that 2023 was one of the weakest NAM years on record, with precipitation anomalies of −1.4 mm/day (Fig. 4a). A correlation analysis between the NAM index and maximum July temperatures over the study region for the 1979–2022 period reveals a negative correlation $r = -0.58$ ($p < 0.01$) and a spatial pattern similar to the 2023 temperature anomalies over the southwest U.S. and Mexico (Supplementary Fig. 5).

It is worth noting that a significant portion of the moisture and thus precipitation that feeds into the NAM region originates from two sources: (1) a Pacific source, mostly through the Sea of Cortez, and (2) an Atlantic source via the Caribbean and Great Plains low-level jet system[72,73]. Previous works have shown that warmer tropical Atlantic SSTs weakens the western edge of the North Atlantic Subtropical High, weakening the Caribbean low-level jet[76], thus modulating moisture

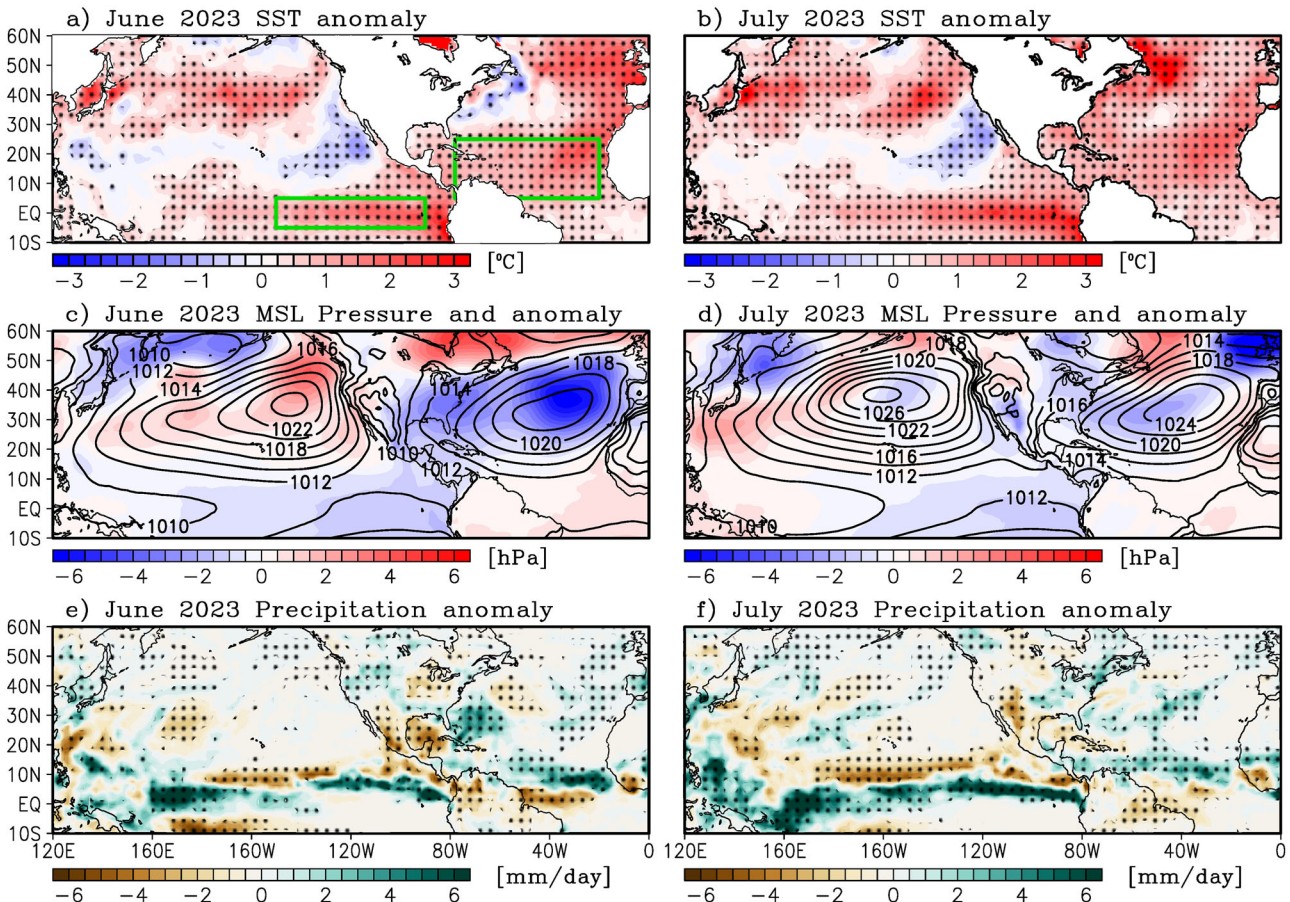

**Fig. 5 | Large-scale circulation anomaly during the boreal summer of 2023. a** Sea surface temperature (SST) anomaly, **c** mean sea level pressure anomaly (color) and total sea level pressure (contour), and **e** precipitation anomaly for June 2023. **b, d, f** are the same as **a, c, e** but for July 2023. Black stipples in **a, b, e, f** indicate statistical significance at a 95% confidence level based on a student $T$ test. Green boxes are the Niño3 [5°N–5°S and 150°W–90°W] and tropical North Atlantic [5°N–25°N and 20°W–80°W] regions.

transport and precipitation over North America[77–79]. As shown in Fig. 4b, the climatological (vertically-integrated) moisture transport into the NAM region originates from the Pacific and Atlantic sectors, converging over the NAM region and thus producing precipitation there. The anomalous moisture transport shown in Fig. 4c represents the sum of the flux of mean moisture by anomalous wind ($v'\bar{q}$), flux of anomalous moisture by mean wind ($\bar{v}q'$), and flux of anomalous moisture by anomalous wind ($v'q'$), where bar (primes) represents climatology (anomaly) respectively, where $v$ is the horizontal wind and $q$ depicts specific humidity. Note that this circulation was disrupted and reversed in 2023 (Fig. 4c), with northerly moisture flux anomalies over the Sea of Cortez and a generally westerly moisture flux anomaly from the Pacific into the Atlantic, all of which resulted in moisture flux divergence over the NAM, reducing the monsoon to one of its weakest on record (Fig. 4a).

To further assess the origin of the reduced NAM and drought conditions, Fig. 4d shows a zonal-vertical cross-section of meridional moisture flux ($vq$) along 28°N from 120°W to 100°W. Similarly, Fig. 4e shows a meridional-vertical cross-section of the zonal moisture flux along 110°W from 15°N to 40°N. These cross-sections are specifically chosen to depict the main core region of the moisture fluxes that feed the NAM. Note that there are two main cores of climatological northward moisture transport (Fig. 4d): (1) around 110°W at about 925 hPa and (2) east of 105°W at higher elevations around 850 hPa. The former is associated with a Pacific moisture source while the latter is linked to an Atlantic moisture source[72,73]. In 2023, significant negative meridional moisture flux anomalies were present over both moisture flux

regions that feed the NAM (Fig. 4d) with significant southward moisture transport and divergence south of 30°N (Fig. 4e). Further analysis on moisture transport anomalies was performed by separating changes in circulation ($v$) and moisture ($q$). It was found that circulation anomalies were primarily responsible for the 2023 reduction in NAM moisture sources and precipitation (Supplementary Fig. 6 and 7). While the reduction (expansion) of the Atlantic (Pacific) subtropical high during the summer of 2023 appeared to be responsible for the reduction in NAM precipitation, other factors like enhanced atmospheric stability from increasing temperatures could have reduced the available surface moisture, inhibiting precipitation and diminishing the NAM[80]. The increased atmospheric stability is exacerbated under uniform SST warming[81], and could have been at play in 2023, given the warm interbasin SSTs.

## Large scale characteristics

Here, we investigate whether any large-scale climate features were responsible for the persistent heat wave. For example, Fig. 5 shows the SST, mean sea level pressure, and precipitation anomalies for June and July 2023 over the Pacific-Atlantic sector. In the tropical Pacific, the positive SST anomalies are indicative of the growing El Niño event, which attained a moderate amplitude by the end of 2023[60]. Further north, the extratropical Pacific SSTs depict a negative anomaly near the west coast of Mexico and California and a positive anomaly along 40°N over the North Pacific current. Over the Atlantic sector, record-breaking warm SST anomalies were observed for the eastern portion of the basin in June, which then migrated westward, encompassing the

entire basin with record SSTs in July. The North Atlantic experienced record warm SSTs in 2023[44], with temperatures greater than 1 °C above the 1981–2010 climatology. Sea level pressure anomalies in June experienced a see-saw pattern with positive anomalies over the Pacific and negative anomalies over the Atlantic, indicating a strengthening of the Pacific subtropical high and a weakening of the Atlantic subtropical high (Fig. 5c). In July, the mean sea level pressure anomalies relaxed over the Pacific and weakened but remained negative over the Atlantic basin. Of interest is the low-pressure anomaly over the southwestern U.S. and Mexico in July (Fig. 5d), indicating the presence of a surface heat low. Precipitation anomalies (Fig. 5e, f) indicate a classical El Niño pattern with a southward shift of the Intertropical Convergence Zone (ITCZ) in the Pacific as well as a reduction in precipitation over the North American Monsoon. Negative precipitation anomalies are also evident over the heat wave region. As depicted in the heat budget analysis (Fig. 2), the surface energy balance showed significant anomalous positive surface sensible heating and negative latent heating during most of the 2023 summer.

Given that both the Atlantic and Pacific basins show remarkable SST anomalies (Fig. 5), it is worth investigating whether these SST anomalies have any connection to the temperature in the southwest U.S. and Mexico. For this, a partial least-square regression analysis is performed using a tropical North Atlantic SST index (TNA, area average over 5°N–25°N and 20°W–80°W) and a tropical Pacific SST index (Niño3, area average over 5°N–5°S and 150°W–90°W), shown by green boxes in Fig. 5a. Although these two SST indices have a temporal correlation of 0.16, and thus are not significantly correlated, we carry out a partial regression analysis to adequately extract their independent linear relationship with respect to 200 hPa stream function, 200 hPa temperature, and 2-m air temperature anomalies (Fig. 6). Note that a combined effect of warm TNA and positive Niño3 (i.e., El Niño) is associated with 200 hPa anticyclonic circulation anomalies throughout the tropics along with 200 hPa warm temperature anomalies that extend over the southern U.S. (Fig. 6c) and corresponding warm 2-m maximum air temperatures (Fig. 6d) that mimic the anomalies associated with the 2023 heat wave event (Fig. 6a, b). Separating the relative contribution of each basin, it is noted that the Atlantic SST anomalies are playing the dominant role given by the regression of TNA-only shown in Fig. 6e, f. Note that the upper-level response is the formation of an anticyclone and warm temperature anomalies over Mexico (Fig. 6e). This is associated with a Gill-type atmospheric response[42] in association with the enhanced diabatic heating over the tropical Atlantic and Caribbean linked to the warm SST anomalies over the TNA[47,79,82]. The 2-m air temperature (Fig. 6f) also shows enhanced warming over Mexico and the southern U.S., similar to the conditions associated with warm TNA and positive Niño3 (Fig. 6d). In contrast, the role of the Pacific SSTs is small, as shown in Fig. 6g, h. Thus, while there was a developing El Niño in the boreal summer of 2023, El Niño events tend to produces tropospheric warming throughout the tropics via a fast equatorial Kelvin wave and off-equatorial anticyclonic anomalies[83–85], similar to Fig. 6e. However, ENSO response to the northern hemisphere is strongest later in the seasonal cycle[83]. Thus, the warm Pacific SSTs appear to have played a minimal role, at least in a linear sense, since extratropical ENSO teleconnections are known to peak in the boreal winter and are relatively weaker in other seasons[83–85]. However, non-linear interactions (i.e., synergy between the warm Atlantic and a growing El Niño) could also be playing an important role. This synergy may not be readily extracted from the simple linear regression presented; thus, a dedicated sensitivity experiment is carried out next.

## Model simulation

The previous sections discussed the coincident climate events (i.e., uniform SST warming from the extreme warm SSTs in the Atlantic, a growing El Niño in the Pacific, and a record low NAM) which could have

been responsible for the occurrence and extended duration of the southwest US and Northern Mexico heat wave in 2023. While attributing a single cause to heat waves is difficult given their synoptic nature, previous works have shown that large-scale climate variations can modulate their occurrence. For instance, precipitation and thus soil moisture are strongly coupled to surface temperatures[74]. It is also known that remote SST anomalies can modulate US climate through circulation changes on interannual timescales[16,86,87] and even on decadal timescales[88]. Therefore, we isolate the effect that the record warm Atlantic SSTs and the growing El Niño had on the longest-lasting southwest US heat wave event of 2023. For this, we perform sensitivity experiments using an atmospheric general circulation model (AGCM, see Methods) by prescribing the observed 2023 SST anomalies over the Atlantic and Pacific basins and climatology elsewhere (Supplementary Fig. 8). Analysis of these AGCM experiments is presented here in terms of the differences between the prescribed global 2023 SST experiment (GBL23) minus the climatology experiment (CTL) (see Methods section). Analysis from the ensemble mean of the 100 ensemble simulations shows enhanced surface net shortwave radiation over northern Mexico and the southwestern US as well as enhanced surface net longwave radiation, enhanced sensible heat flux, and reduced latent heat flux (Supplementary Fig. 9). These surface energy fluxes are consistent with enhanced warming and reduced moisture over the analysis region and consistent with the anomalies observed in 2023.

The large-scale circulation changes associated with 2023 SSTAs are shown by the difference between GBL23 minus CTL experiment for June-July-August (JJA) 200hPa temperature and 200 hPa streamfunction (Fig. 7a). Note the anomalous heat dome structure and associated anticyclonic circulation pattern over Mexico and the southwestern US, which is remarkably consistent with the observed partial regression pattern (Fig. 6c). Here, the model responds to the 2023 SST forcing by producing an anticyclonic circulation anomaly over Mexico (positive contours), leading to enhanced near surface temperatures for most of the region (Fig. 7b). While the AGCM response is very similar to that of the observed analysis (Fig. 6), it is worth separating the relative contributions of Atlantic and Pacific SST anomalies. For this, we look at the SST sensitivity experiments with Atlantic-only (ATL23) and Pacific-only (PAC23) prescribed 2023 SSTs. Consistent with the regression analysis, the Atlantic SST forcing appears to play a more dominant role by creating an upper-level anticyclonic circulation anomaly (Fig. 7c) and enhanced near-surface warm temperatures (Fig. 7d) comparable in amplitude to that of the GBL23 experiment. In contrast, the PAC23 experiment shows relatively weak homogeneous upper tropospheric warming and anticyclonic circulation at low latitudes (Fig. 7e), a feature typical of a developing El Niño event. Thus, the surface temperature signal over the southwestern U.S. and Mexico is very small in comparison (Fig. 7f).

The sources for these teleconnections are analyzed via Rossby wave sources (RWS, see Methods). Note that there are a few regions of anticyclonic RWS in the GBL23 experiment (Fig. 8a) over the equatorial Pacific and Atlantic, near the location of the ITCZ, and also over the Greater Antilles. These are consistent with the warm 2023 SST forcing prescribed in the GBL23 experiment, which led to upper-level 200 hPa divergence (Fig. 8c) and 200 hPa anticyclonic circulation consistent with a Gill-type response (Fig. 8b). Further decomposition of the RWS into its component demonstrates that mean vortex stretching by the anomalous divergent flow (Fig. 8e) was the main driver of the anticyclonic RWS and thus the anticyclonic circulation over Mexico.

The individual roles of each ocean basin are investigated using the ATL23 and PAC23 sensitivity experiments. For the Atlantic-only forcing (Supplementary Fig. 10), a similar pattern emerges where the upper-level divergent flow being responsible for the anticyclonic RWS over the Atlantic basin, which leads to a downstream (westward) intensification of the upper-level anticyclone over Mexico and the eastern

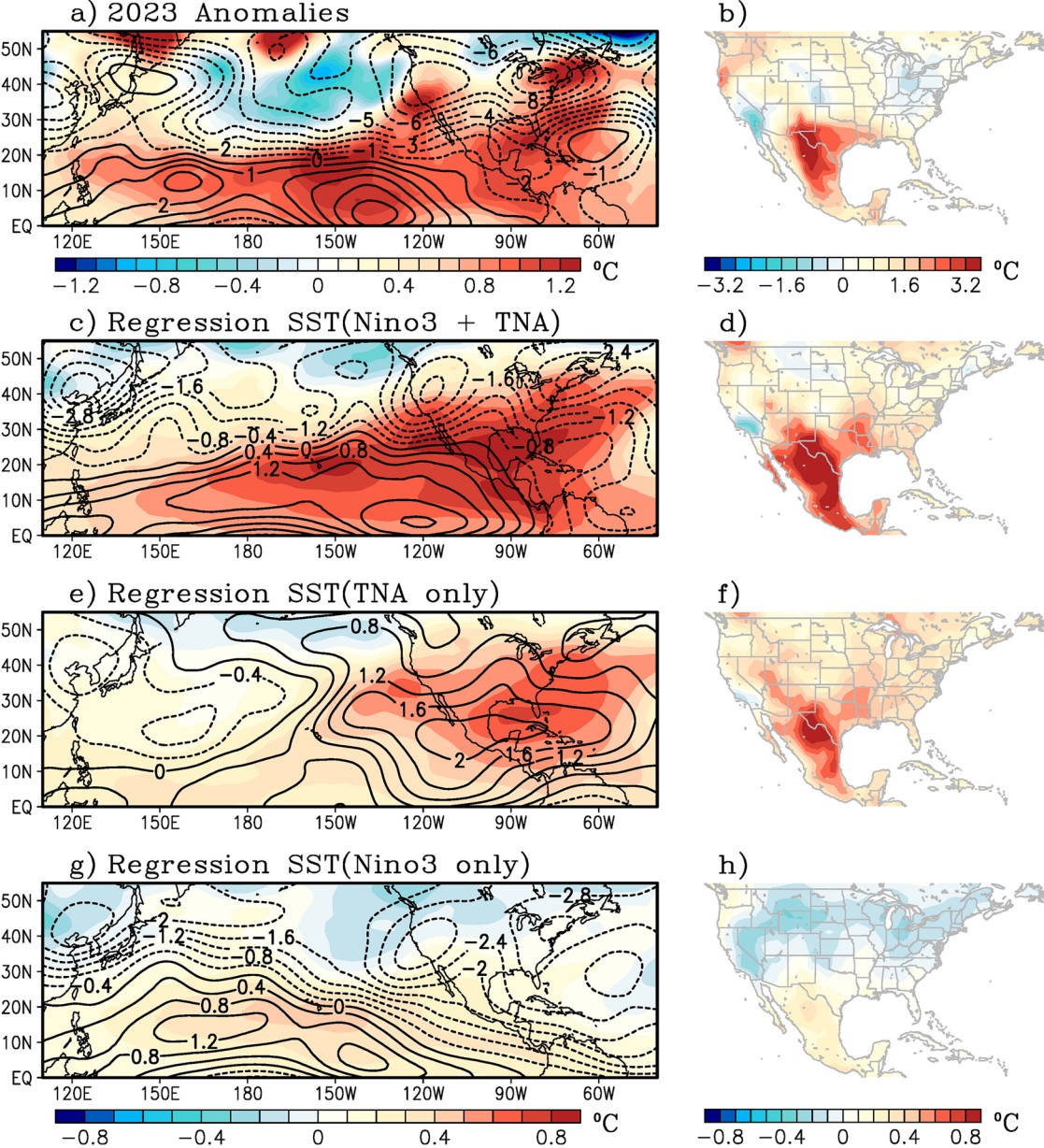

**Fig. 6 | Observed linear relationship between tropical sea surface temperature (SST) and atmospheric circulation. a** Anomalous 200 hPa temperature (color) and streamfunction (black contour, $10^6 \, s^{-1}$) for June–August of 2023. **b** same as (**a**) but for 2-m maximum air temperature anomaly. **c** Regression coefficient of Niño3 plus tropical North Atlantic (TNA) SST and 200 hPa temperature (color) and 200 hPa streamfunction (black contour, $10^6 \, s^{-1}$). **d** Same as (**c**) but for 2-m maximum air temperatures. The regression coefficients are computed for June–August for the 1979–2023 period through partial regression, and the units are per standard deviation of the SST anomalies. Panels (**e**, **f**) are the same as (**b**, **c**) but for the regression coefficients with TNA SSTs only and holding Niño3 SSTs constant. Similarly, (**g**, **h**) show the regression coefficients with respect to Niño3 SSTs, holding TNA SSTs constant.

Pacific between 20 and 40°N. In contrast, the Pacific-only SST sensitivity experiment (Supplementary Fig. 11) yields anticyclonic RWS over the ITCZ, in association with the developing El Niño, and consistent with upper-level divergence over the heating region[83]. However, this signal is smaller than those from the ATL23 forcing. The larger RWS from the ATL23 compared to the PAC23 sensitivity experiment is probably due to the fact that SSTs in the Atlantic were at their peak while the El Niño in the Pacific was still in the growing phase (Supplementary Fig. 8).

As discussed earlier, non-linear interactions (i.e., synergy between the warm Atlantic and a growing El Niño) could also be playing an important role in exacerbating extreme heat in the region. This interaction can be readily extracted from the AGCM experiments by isolating the interbasin synergy following Eq. 7 (see Methods) and is

shown in Fig. 7g, h). Note that the synergy component mostly shows upper tropospheric warming and relatively warm near surface temperatures over the southwest US, while troughing and cooler temperatures upstream and downstream of the heat wave region (Fig. 7h).

Besides changes in large-scale circulation patterns, the occurrence of heat wave events is also investigated within the AGCM experiments. For this, daily outputs of maximum temperature from the model experiment are used to compute changes in several heat wave characteristics. Here, the daily T95 percentile maximum temperature is computed from a 300-year control simulation. Then, heat wave amplitude, number of heat wave events, number of heat wave days, and the duration of each event are computed relative to the control T95 (Fig. 9). For all four indices, the difference between the GBL23 and CTL experiment is normalized by the expected occurrences

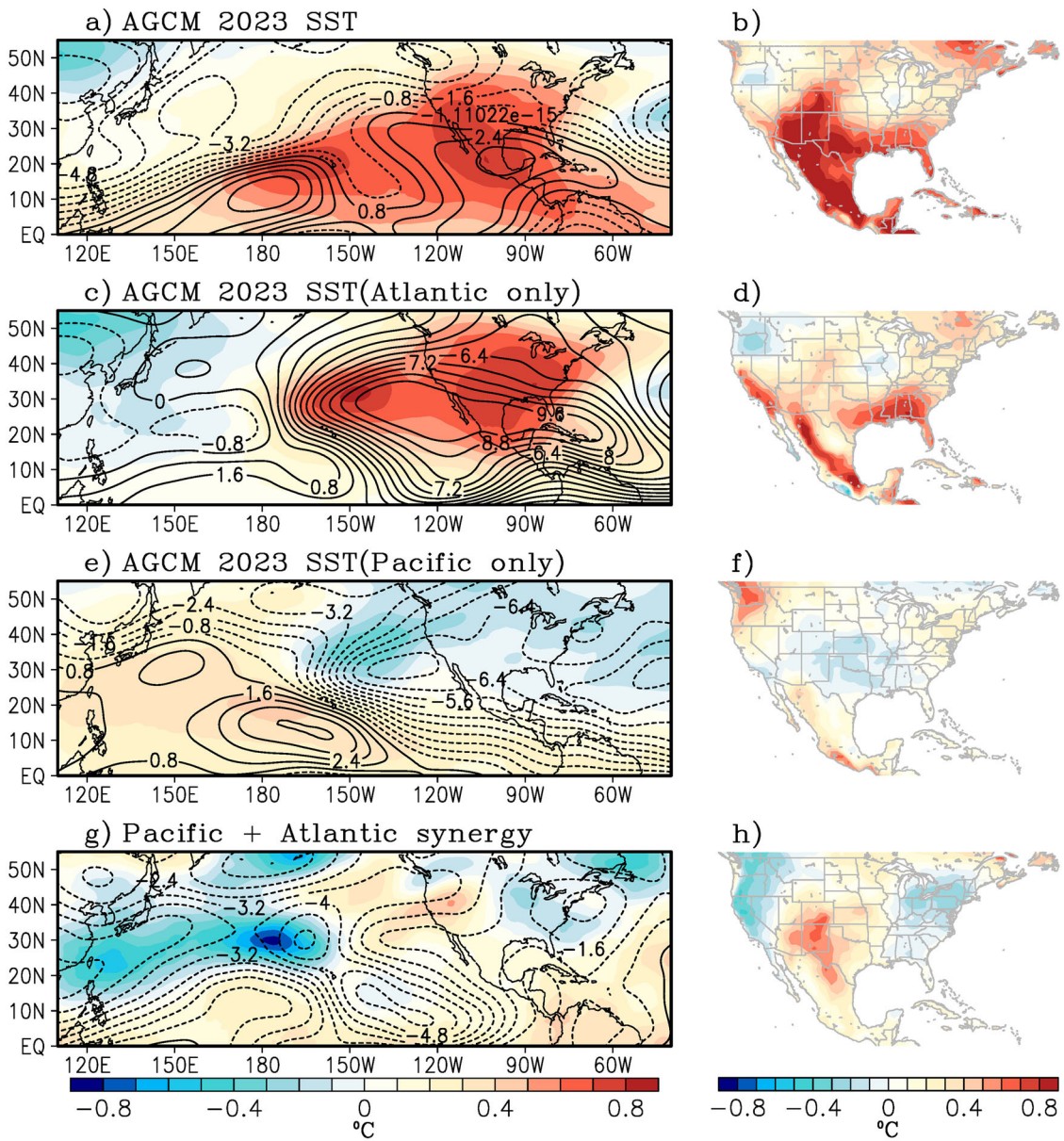

**Fig. 7 | Sea surface temperature (SST) sensitivity from an atmospheric model experiment.** Composite difference of simulated (**a**) 200 hPa temperature (color) and 200 hPa streamfunction (black contour, $10^6\,s^{-1}$) and **b** 2-m air temperature from the atmospheric general circulation model (AGCM) experiment with prescribed 2023 global SSTs (GBL23). **c**, **d** are the same as (**a**, **b**) but for the AGCM experiments with Atlantic-only 2023 SSTs (ATL23). Similarly, (**e**, **f**) show the composites from the AGCM experiment with prescribed Pacific-only SSTs (PAC23). Similarly, (**g**, **h**) show the synergy between the Atlantic and Pacific forcings (see Methods). The composite differences are with respect to the control experiment (CTL) for JJA.

in the CTL experiment, e.g., (GBL23 – CTL)/CTL. The relative changes between the GBL23 relative to the CTL experiment show an amplitude increase on the order of 10% (mostly over Mexico, where amplitude is measured as the maximum 2-meter temperature anomaly averaged over the duration of the event). The number of heat wave events more than doubled (e.g., greater than 100% increase), the number heat wave days tripled, and a >20% increase in the duration of heat waves occurred over the southwestern US and Mexico in the GBL23 sensitivity experiment. Thus, the sensitivity experiment further validates the role of the 2023 SSTs in modulating the occurrence and persistence of heat wave events over the study region.

A composite analysis for several heat wave metrics is shown in Fig. 10 for the grid point closest to Phoenix, Arizona from the AGCM experiment. For instance, the CTL (GBL23) case produced 77 (184) heat wave events. This is a 138% increase over the expected climatology, of

which 75% is attributed to Atlantic SSTs, 31% to the Pacific, and 32% to non-linear or synergic interbasin interactions (Fig. 10a). Similarly, a total of 307 (853) heat wave days were produced by the CTL (GBL23) experiment (Fig. 10b), a 177% increase above climatology of which 89%, 27%, and 61% are attributed to the Atlantic-only, Pacific-only, and interbasin SST synergy respectively. The average event duration also increased by around 18% from the climatological 3.9 days to 4.6 days (Fig. 10c), of which 10% was due to the Atlantic-only and the remaining 8% from the interbasin synergy, and no contribution from the Pacific-only case. The longest-lasting heat wave in the CTL (GBL23) experiment was 9 (14) days, all while the GBL23 experiment experienced several double-digit event durations (Supplementary Table 1). Besides more and longer-lasting events, the GBL23 experiment also produced larger amplitude events. Looking at the return period in years of a very high threshold (a high threshold is chosen here as a temperature

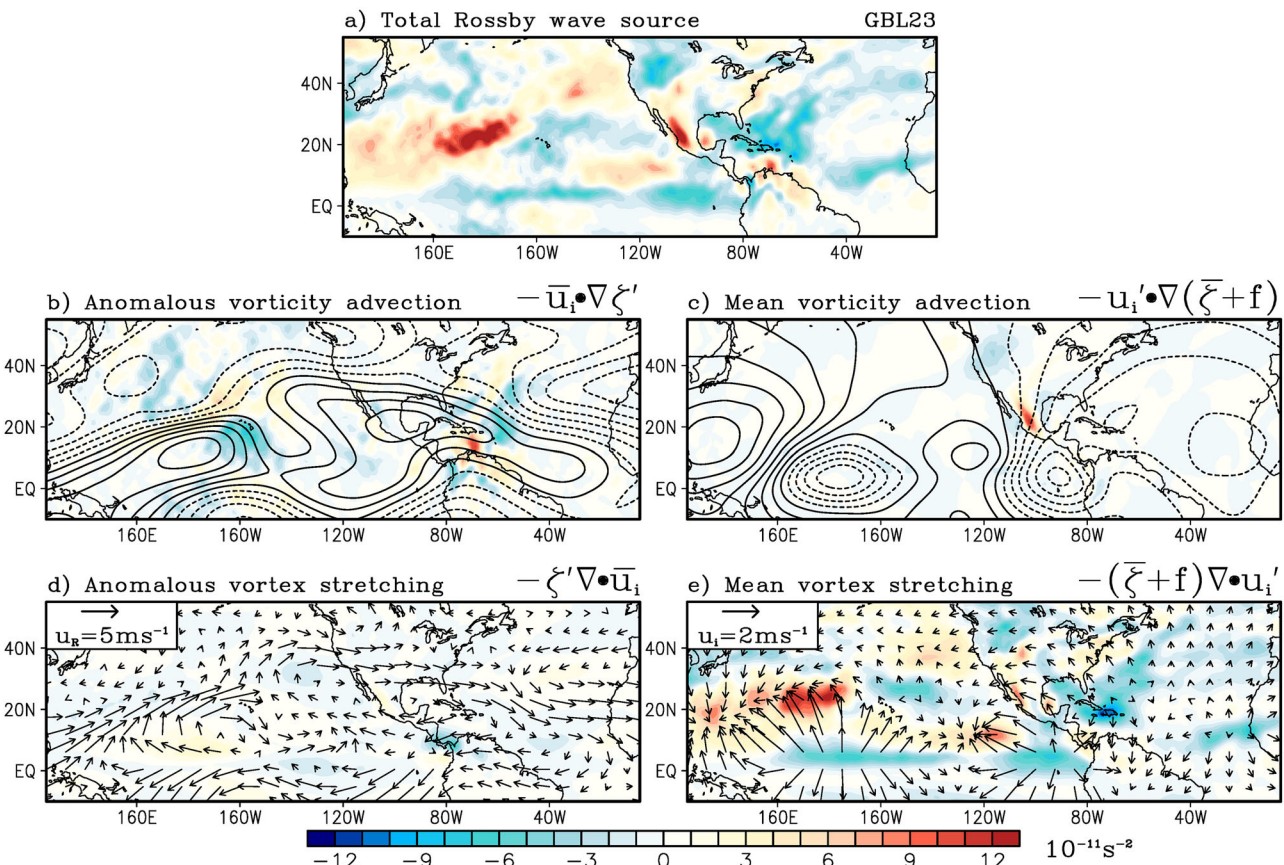

**Fig. 8 | Sea surface temperature (SST) sensitivity from the atmospheric model experiment with prescribed global 2023 SSTs (GBL23).** Composite difference of simulated (**a**) 200 hPa Rossby wave source (RWS), **b** 200 hPa streamfunction (contour, $10^6\,s^{-1}$) and anomalous vorticity advection (color), **c** 200 hPa velocity potential (contour, $10^6\,s^{-1}$) and mean vorticity advection (color), **d** 200 hPa rotational wind (vector, $ms^{-1}$) and anomalous vortex stretching (color), and **e** 200 hPa divergent wind component (vector, $ms^{-1}$) and mean vortex stretching. The units for the RWS terms are $10^{-11}\,s^{-2}$, see Methods for definition. Composites are from the atmospheric general circulation model (AGCM) experiment with prescribed 2023 global SSTs (GBL23). The composite differences are with respect to the control experiment (CTL) for June-July-August.

similar to the 2023 heat wave event) and modeled by a Pareto distribution[89], a 45 °C daily maximum temperature is observed once every 45 (14) years in the CTL (GBL23) experiment (Fig. 10d). This represents a 222% increase in frequency above climatology (e.g., frequency is inversely proportional to return period), where 137%, 60%, and 25% were due to the Atlantic-only, Pacific-only, and interbasin SST synergy respectively.

The JJA daily maximum temperature is also significantly higher in the GBL23 experiment (ensemble mean of 40.2 ± 0.31 °C) compared to the CTL experiment (ensemble mean of 39.1 ± 0.30 °C), see Supplementary Table 1. This difference of over 1 °C is significant against the background weather noise, which is taken from the ensemble spreads of the 100 realizations from each experiment. Similarly, the minimum temperature is also higher in the GBL23 experiment (26.8 ± 0.34 °C) compared to the CTL case (25.8 ± 0.31 °C). A daily minimum temperature excess over a very high threshold (here 31 °C) is observed every 30 (11) years in the CTL (GBL23) experiments (Supplementary Table 1).

## Discussion
This work investigated the record-breaking, large amplitude, broad spatial scale, and persistent heat wave event that impacted the southwestern U.S. and Mexico in 2023. This heat wave set multiple warm temperature records, both for the maximum and minimum daily temperatures in multiple locations. It was also responsible for the longest stretch of very high temperatures in multiple cities across

the U.S. and Mexico, which contributed significant stress on human health, agriculture, and infrastructure. It was found that a large-scale, semi-persistent atmospheric blocking pattern was anchored over the region for most of the summer, a feature that enhanced incoming solar radiation, thermal heating, and sensible heating, deep subsidence, while reducing precipitation, thus increasing surface temperatures. The atypical features of this extreme event suggested that there were slow-varying large-scale forcings at play, potentially promoting its occurrence. It was shown that the homogeneous interbasin warm SSTs from the record warm Atlantic Ocean and a growing El Niño event in the Pacific modulated large-scale atmospheric circulation. These circulation changes thus promoted a semi-permanent anticyclonic blocking pattern, significantly increasing surface heating, which led to the growth and persistence of the heat wave. In addition, the NAM was one of the weakest on record, as a result of reduced moisture fluxes from the Pacific and Atlantic sectors. A partial regression analysis from observations and dedicated model experiments with prescribed SSTs confirmed that the extremely warm tropical Atlantic Ocean in 2023 was the dominant factor, which increased the likelihood of heat waves over the region. Meanwhile, the Pacific SSTs' influence was much smaller, at least when measured as a stand-alone forcing. However, the interbasin synergy effect of Pacific-Atlantic forcing proved to be central in further exacerbating the likelihood of heat waves in the study region, including extending their duration and enhancing the warm temperature anomalies.

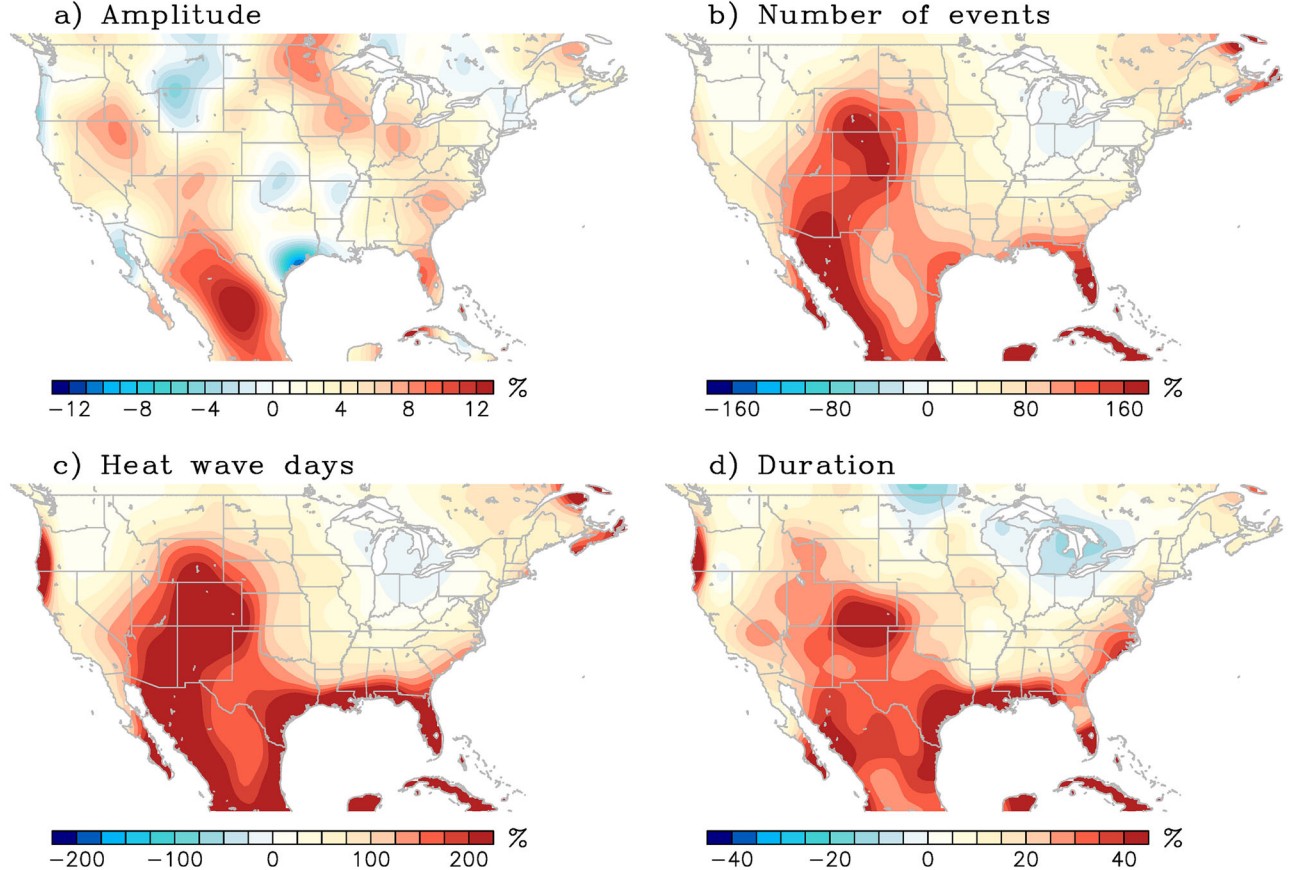

**Fig. 9 | Heat wave response to imposed sea surface temperature forcing.** Composite difference of simulated 2023 minus control atmospheric general circulation model (AGCM) experiments during June-July-August for (**a**) average amplitude of heat waves, **b** number of heat wave events, **c** number of heat wave days, and **d** heat wave duration. Units are percentage changes as defined by the number of events during the simulated 2023 minus control AGCM divided by the total events in the control experiment, and multiplied by 100, such that a 100% increase translates into a doubling of the occurrence.

There are many factors potentially contributing to the record warm Atlantic SSTs. On the global scale, the ocean continues to warm, not only at the surface but at depths as well[44], with ocean heat content steadily increasing due to the Earth's energy imbalance[48,49]. However, the warming has been concentrated in the upper ocean, with the effect of increasing stratification[44,90]. This makes the upper ocean more stable and less prone to mixing by the winds, exacerbating the surface warming, and thus increasing the SSTs further. This is very relevant for the North Atlantic, where the current warming has been shown to be concentrated near the surface[44,90]. On decadal timescales, the Atlantic Multidecadal Variability[46,47], also influences the enhanced ocean heat content. On shorter timescales, El Niño developed in 2023, which has been shown to redistribute heat from deeper layers to those near the surface, thus also yielding higher SSTs[91,92]. Locally in the Atlantic, the winds were weaker than normal due to a series of atmospheric features. A weaker Bermuda High, which induces a weaker near-surface wind, enhances warming through reduced evaporative cooling at the surface[93,94]. Other factors like enhanced shortwave solar radiation and reduced aerosols could also be at play[90,95,96], including emission regulations from the International Maritime Organization[53,54]. However, causes for the extreme warm Atlantic SSTs are a subject of further investigation, outside the main scope of this paper.

## Methods
### Observational data
Atmospheric variables (e.g., vertical profiles of temperature, moisture, geopotential height, wind, and heating rates) are obtained from

monthly and daily means from the Japanese 55-year Reanalysis[97] (JRA55) for the period of 1955–2023 with a 55 km spatial resolution, and 60 vertical levels up to 0.1 hPa. Data interpolated to 1.25° × 1.25° spatial resolution and 37 vertical levels from 1000 hPa to 1 hPa were used. Heating rates are provided directly by the JRA55 reanalysis, which includes net atmospheric radiation (short and longwave radiation), vertical diffusion heating rate, which represents the energy transfer without phase change; latent heat from moist processes, which includes the large-scale condensation heating rate and the convective heating rate. Daily maximum and minimum temperatures are obtained from meteorological observations for several stations along the southwestern U.S. from the NOAA/National Center for Environmental Information from 1950 to 2023. Observed SSTs are obtained from the Hadley Centre HadSSTv2 product[98] at a 1° horizontal resolution for the period of 1900–2023. Anomalies are defined relative to the climatological period from 1979 to 2022.

### Heat wave definition
We assess heat waves by the Excess Heat Factor[99] (EHF) used in operational forecasts as well as in research studies[15,16,100]. The *EHF* index (Eq. 1) is defined by combining two excess heat indices, namely an acclimatization index (Eq. 2), which measures the current 3-day temperature anomaly relative to the previous 30-day period; and a significant exceedance index, which measures the excess 3-day temperature over a high threshold, taken here as the 95th percentile temperature for that given day (Eq. 3). Here, $T_i$ corresponds to the daily maximum surface temperature for day $i$ and $T_{95}$ is the 95th percentile temperature for that given day. A positive *EHF* characterizes heat wave

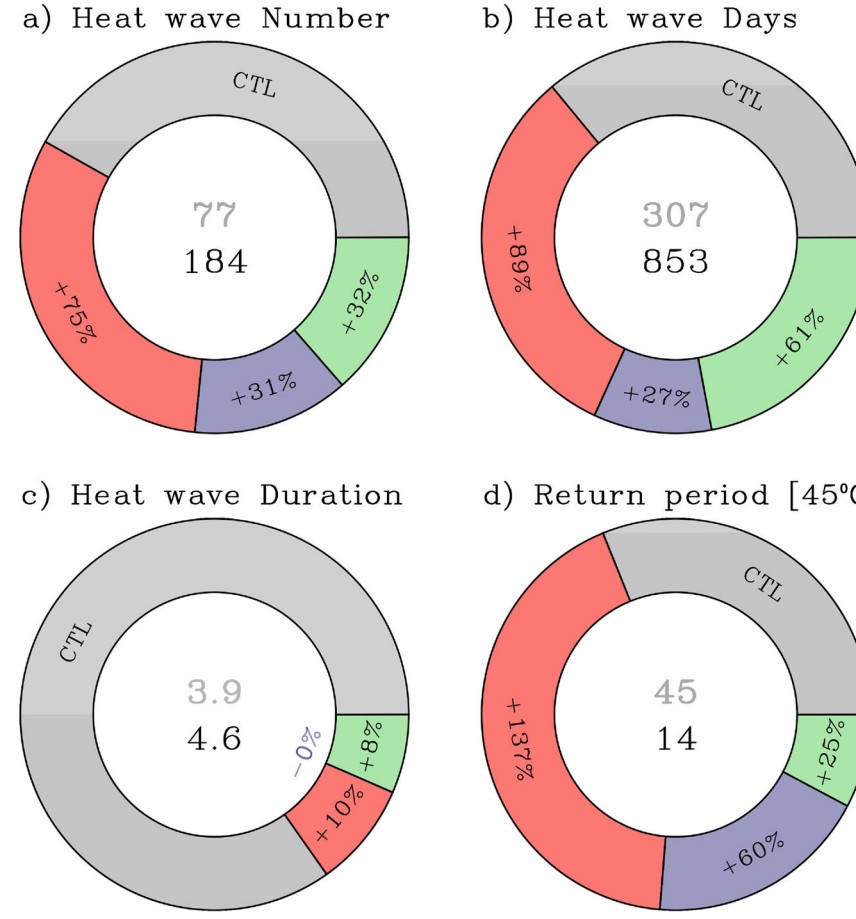

CTL    **ATL23**    PAC23    Synergy

**Fig. 10 | Relative contribution of several components to the occurrence of heat waves. a** Heat wave number, **b** heat wave days, **c** duration in days, and **d** frequency of exceeding a 45 °C threshold in years for the grid-point closest to Phoenix, Arizona. The components are: the expected value or climatology (CTL), Atlantic-only 2023 (ATL23) sea surface temperature (SST) sensitivity, Pacific-only 2023 SST sensitivity (PAC23), and the Pacific-Atlantic Synergy, scaled by the total contribution corresponding to the GBL23 experiment (2023 SST sensitivity). The numeric values in the center are the climatology (top) and 2023 SST sensitivity experiment (bottom). The percentage values in each of the slices indicate the increase/decrease relative to climatology (CTL, or 100%). These values are from the atmospheric general circulation model AGCM experiments. See Methods.

conditions that persist for a minimum of three days.

$$EHF = \max\left[1, EHI(accl.)\right] \times EHI(sig.) \quad (1)$$

$$EHI(accl.) = \frac{(T_i + T_{i-1} + T_{i-2})}{3} - \frac{(T_{i-3} + \ldots + T_{i-32})}{30} \quad (2)$$

$$EHI(sig.) = \frac{(T_i + T_{i-1} + T_{i-2})}{3} - T_{95} \quad (3)$$

**Heat budget**

To isolate the physical processes responsible for the development and long persistence of the heat wave, a heat budget analysis is performed (Eq. 4) for the temporal evolution of a temperature anomaly $T$ due to horizontal advection $(\vec{u} \cdot \nabla T)$, vertical advection $\left(\omega \frac{\partial T}{\partial p}\right)$, adiabatic heating $\left(\omega \frac{RT}{C_p P}\right)$, and diabatic heating ($Q_{net}$). This last term consists of shortwave, longwave, diffusive, and latent heating. In Eq. 4, $\vec{u}$ and $\omega$ are the horizontal and pressure velocities, $R = 287\,\mathrm{J\,Kg^{-1}\,K^{-1}}$ is the ideal gas constant for dry air, and $C_p = 1,004\,\mathrm{J\,Kg^{-1}\,K^{-1}}$ is the specific heat of dry air at constant pressure. Equation 4 is vertically integrated from

pressure level $975 - 800$ hPa, chosen here to be representative of low-level temperature variations[101]. The integral is mass-weighted and normalized by the total integration thickness $\Delta P$ to preserve the original units of °C/day of the heating rates and for easier comparison.

$$\frac{1}{\Delta P}\int_{975}^{800}\frac{\partial T}{\partial t}\,dp = \frac{1}{\Delta P}\int_{975}^{800}\left[-\vec{u}\cdot\nabla T - \omega\left(\frac{\partial T}{\partial p} - \frac{RT}{C_p P}\right) + Q_{net}\right]dp \quad (4)$$

**Trapped flow characteristics**

Flow characteristics associated with blocking pattern and heat wave dome are assessed by the instantaneous local Lyapunov exponent (Eq. 5).

$$\lambda_+ = \frac{1}{2}\left[D + \left(E^2 - \zeta^2\right)^{1/2}\right] \quad (5)$$

Where, $\lambda_+$ is the positive Lyapunov exponent, which is equivalent to the dilation rate. Here, $D = \partial u/\partial x + \partial u/\partial y$ is the horizontal divergence, $E = \sqrt{E_{st}^2 + E_{sh}^2}$ is the deformation, $E_{st} = \partial u/\partial x - \partial v/\partial y$ is the stretching deformation and $E_{sh} = \partial u/\partial y - \partial v/\partial x$ is the shear deformation, and

$\zeta = \partial v/\partial x - \partial u/\partial y$ is the relative vorticity. The imaginary component of $\lambda_+$ (i.e., where $\zeta = \partial v/\partial x - \partial u/\partial y$) is used to represent regions of fluid trapping[102], or where the flow is dominated by vorticity. If $\zeta < 0$, then the trapped flow is anticyclonic in nature, thus leading to significant air mass modification due to reduction in ventilation from adjacent air masses, and reinforcing the heat dome and surface warming[103].

### North American Monsoon (NAM) definition

The NAM index follows the concept of the global monsoon[104,105], defined by area-averaging precipitation over western North America with the constraints that the June-July-August-September minus December-January-February-March precipitation range is greater than 2 mm/day and the local summer precipitation exceeds 55% of the total annual precipitation.

### Rossby wave source

Rossby wave source[106] is defined by Eq. 6, where the anomalous RWS is defined by the anomalous vorticity advection by the mean divergent wind, mean vorticity advection by the anomalous divergent winds, anomalous vortex stretching, and mean vortex stretching, respectively. Here, $u$ is the divergent wind (i.e., irrotational component of the flow), $\zeta$ is the relative vorticity, and $f$ is the Coriolis parameter. Overbar denotes the mean climatology, and primes represent the deviation from climatology.

$$RWS = -\nabla(u_i\zeta)' = -\bar{u} \cdot \nabla\zeta' - u' \cdot \nabla(\bar{\zeta} + f) - \zeta'\nabla \cdot \bar{u} - (\bar{\zeta} + f)\nabla \cdot u' \tag{6}$$

### Model experiments

To isolate the effect of significant tropical SST warming, several atmospheric general circulation model (AGCM) experiments are performed. We prescribe SST anomalies to the Community Atmosphere Model version 6 (CAM6) coupled to the Community Land Model version 5 (CLM5), which are part of the Community Earth System Model version 2 (CESM2). First, the control case is integrated by prescribing the SST climatology globally based on the 1979-2022 observing period, which is referred to here as the CTL case and is run for 300 years. Then, a model experiment is conducted where SST is prescribed based on the observed 2023 SST only over the Atlantic and Pacific and is held to climatology elsewhere, referred here onward as the GBL23 case (see Supplementary Fig. 8 for prescribed SST anomalies). For the GBL23 case, the last 100 years of the CTL are used as initial conditions to create 100 GBL23 ensembles. That is, the experiments are integrated for one year starting on every 1 January from the CTL (e.g., branch simulation), thus creating ensembles with initial conditions that are one year apart from each other. This approach guarantees more than sufficient separation among ensembles as it pertains to weather noise (i.e., the atmospheric initial state is completely different and independent among ensembles) while each ensemble is being forced by the same underlying SST anomalies.

To isolate the relative influences of the Atlantic and Pacific SST anomalies on atmospheric circulation, two additional model experiments are performed where the Atlantic-only and Pacific-only 2023 SST anomalies are prescribed, these experiments are referred to as ATL23 and PAC23, respectively. All experiments were integrated under a year-2000 atmosphere composition (i.e., CESM2 component set F2000). The experiments are evaluated with respect to their differences relative to the CTL runs, thus, any contrast between the two experiments is attributed to SST anomalies in 2023, whereas the ensemble spread of each model run is used as uncertainty estimation (i.e., weather noise) through a bootstrapping technique. It is worth mentioning that besides the modulating effects of circulation, heat waves are also influenced by complex land-atmosphere interactions/feedbacks that may not be well resolved in CESM2. In addition, significant topographic features are present in the study area, which have been shown to play an important role in surface temperature variations, but may not be well-resolved at these horizontal resolutions. Moreover, complex air-sea interactions are also missing from these AGCM simulations as the atmospheric model is forced by SSTs with no ocean thermodynamic nor dynamical feedback.

### Atlantic-Pacific synergy (non-linear interactions)

The synergy between the Atlantic and Pacific SST sensitivity is extracted from the AGCM experiments following Eq. 7. This is possible because the GBL23 experiment comprises the total linear plus non-linear SST sensitivities, whereas the targeted basin experiments (i.e., ATL23 and PAC23) isolate interbasin interactions. See Supplementary Information for more details on the definition of interbasin synergy and derivation of Eq. 7.

$$Interbasin\ Synergy = GBL23 - ATL23 - PAC23 + CTL \tag{7}$$

### Statistical significance—bootstrapping technique

A bootstrapping method is used to determine confidence intervals by subsampling the dataset. All analyses presented are obtained by randomly selecting $r$ samples out of $n$ observations with replacement (Eq. 8). This is done 500 times in order to build a significant distribution of composites and assign 95th percentile confidence levels.

$$(n\ r) = \frac{n!}{r!\,(n-r)\,!} = possible\ combinations \tag{8}$$

## Data availability

Hadley Centre HadSSTv2 product for the period of 1900–2023 was obtained from https://www.metoffice.gov.uk/hadobs/hadisst/ (Rayner et al.[98]). The Japanese 55-year Reanalysis was obtained from https://rda.ucar.edu/datasets/ds628.0/ (JRA55, Kobayashi et al. [97]) for the period of 1955 – 2023. Daily maximum temperatures are obtained for meteorological observations for Phoenix, Arizona from the NOAA/National Climate for Environmental Information from 1950 – 2023 https://www.ncdc.noaa.gov/cdo-web/search?datasetid=GHCND. Source data are provided with this paper.

## Code availability

Details on installing and running CESM2 can be found at https://github.com/ESCOMP/CESM. The data in this study were analyzed using the publicly available Gridded Analysis and Display System (GrADS; http://cola.gmu.edu/grads/).

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

## Acknowledgements

This research was carried out in part under the auspices of the Cooperative Institute for Marine and Atmospheric Studies, a cooperative institute of the University of Miami and the National Oceanic and Atmospheric Administration (NOAA), cooperative agreement NA 20OAR4320472. Hosmay Lopez and Sang-Ki Lee acknowledge support from base funds from AOML's Physical Oceanography Division.

## Author contributions

H.L. conceived the study and wrote the initial draft of the paper. H.L., S.K.L., R.W., D.K. and L.J. contributed to the design, the statistical analysis, and interpretation of the results as well as the writing of the final version of the paper.

## Competing interests

The authors declare no competing interests.
