## [Transparent Peer Review file · Nature Communications]

The longest-lasting 2023 western North American heat wave was fueled by the record-warm Atlantic Ocean

Corresponding Author: Dr Hosmay Lopez

Version 0:

Reviewer comments:

Reviewer #1

(Remarks to the Author)
The manuscript

"The longest-lasting 2023 western North American heat wave was fueled by the record-warm Atlantic Ocean" by Hosmay Lopez, Sang-Ki Lee, Robert West, Dongmin Kim, Liwei Jia

features an in-depth investigation of the causes and processes related to the outstanding heatwave that affected southwestern US and Mexico between June and July 2023. By analysing in-situ measurements, remote sensing, gridded observations and reanalyses, the authors argue that the event in the region of interest, and particularly in the area of Phoenix, was characterized by abnormal positive anomaly of sensible, diffusive, and radiative surface fluxes towards the surface. They also found that an anticyclonic pattern, suppressing advective fluxes and enhancing local adiabatic heating, were responsible for the long heat streak across much of July. The North American Monsoon, and moisture fluxes entering the area from the oceans, were anomalously weak. Regression analyses evidenced that the role of North Atlantic SSTs was more relevant than that associated with subtropical Pacific SSTs. This was confirmed by sensitivity analyses using a General Circulation Model in AMIP settings, forcing the atmosphere with prescribed SSTs representative either of the overall SST anomalies in Summer 2023, or of the anomalies over the Atlantic and over the Pacific during the same period. The interbasin feedback is claimed to play a substantial role in the build-up of the anomaly, therefore the authors stress that the action of both basins has to be taken into account when attempting to attribute the North American heatwaves to anthropogenic climate change enhanced by oceanic natural variability.

Overall, I think that the manuscript has scientific merits, and is worth publication. However, there are several substantial aspects that should be carefully taken into account before the manuscript can be accepted.

My main concern is that in the section involving observations authors fall a bit short in justifying the correlations of different aspects of the circulation with process understanding. Another aspect that should be clarified is the attribution of extreme events to the role of Atlantic-Pacific-synergy of both SSTs. Therefore, I recommend the manuscript for major revision.

Specific comments are detailed below.

- I. 27: 2024 surpassed 2023 as warmest year on record. Please mention it.
- I. 36: Is there any reason why Phoenix would be an exceptional spot where to focus the analysis? The rationale of the work would be certainly improved by providing a justification for this;
- II. 43-45: I see that all these examples mention Northern Hemisphere continents, but outstanding heatwaves also affected the Southern Hemisphere. Perhaps worth mentioning it.
- II. 53-54: it might be worth mentioning that the effects of heatwaves are also exacerbated by increasing urbanization, agricultural loss, increased aridification;
- I. 75: furnished -> provided
- I. 77: I may agree that the oceanic timescales have timescales that are longer than those pertaining on local heat extremes, but I am not sure this applies to land process variabilities as well (e.g. Hebert et al. 2022);
- II. 82-83: Perhaps it is worth mentioning Kuhlbrodt et al. 2023 on the role of ocean heat content for North Atlantic 2023 extreme;
- I. 97: it is surprising that the authors seem to be unaware of the active debate on the interbasin Atlantic-Pacific

teleconnections, especially in the Tropics, that has developed in years. I am not in the position to provide a conclusive literature survey, but I may suggest that the authors have a look at the debate around the Jiang et al. 2023 paper (e.g. Richter et al. 2024) and provide some references in the manuscript.

- I. 101: related to my previous comment, it could be better emphasized that the findings that are actually contributing substantially to the topic regard the inter-basin synergy. So far, it seems to the reader that is just one of the many aspects touched here in the context of the 2023 heatwave;
- I. 104: it shall be mentioned explicitly here what is the reference period for the computation of the climatology;
- I. 109: I struggle a bit to read Figure 1. I wonder why the authors prefer showing absolute temperatures instead of anomalies, that would much better emphasize the T95 exceedances;
- I. 129: Figure 2 is very chaotic in my opinion, and I think it shall be improved by showing actual time series, instead of boxes with 10-days averages. If the daily spread is too high, a 10-days moving average could be superimposed on the actual time series. Also, I think that it is very misleading to position color bars next to x-axes, whose labels are only found at the bottom of the figure;
- I. 140: Figure 3c may be obvious to a reader with some sort of meteorology background, but it is not at all for many readers of Nature Communications. Therefore, the skew-T should be carefully explained (perhaps in the Methods?) making a bit more explicit what the authors discuss, particularly at lines 184-188 and in Figure 3c description;
- I. 142: it is nowhere explained what the authors mean by "diffusive heating" and how this is computed;
- I. 146: similarly to my comments above: it is not worth mentioning that the Bowen ratio increases if it is not explained what this means in terms of process understanding;
- I. 159: wouldn't it be clearer to state that the dynamical tropopause crosses the 2 PVU isosurface where the thick solid line is?
- I. 165: not clear why I would infer from Figure 2 that there is shortwave surface heating in clear sky, i.e. why I would have to assume clear sky;
- I. 168: it is textbook literature that atmospheric blocking is among the most prominent features of extratropical low frequency variability (e.g. Rex et al. 1950), then some sort of agreement shall be reached on what the authors refer to as "synoptic", because blocking is not proper of synoptic timescales (although it might be to some extent of synoptic spatial scales, although I would rather refer to it as a manifestation of planetary waves; cfr. Stoll et al. 2023 and references therein)
- I. 198: what is the exact definition of "western North America" here?
- I. 202: this is another example of one of my main concerns about (this section of) the manuscript. Besides showing that the North American Monsoon is weak in 2023, claiming that some sort of correlation is present between moisture fluxes enter the region and negative latent heat fluxes locally (in Phoenix, at least), what do we learn about the relation between exceptional heatwaves in the region and the monsoon? Could it be just a chance? What happened in 1982 (the weakest NAM on records, according to the time series in Figure 4a)? Is the anomaly significant wrt. NAM climatology?
- I. 211: related to my previous comment; authors are comparing the NAM activity along the whole season, but the heatwave peaked during 14 days in July and short heatwave periods emerged during August and September (possibly not related to the major one). It would perhaps be more informative if the evolution of daily moisture transports could be tracked during the season;
- I. 254: how relevant is negative anomaly (shown in mm/day) compared to the precipitation that usually occurs during July in the region (i.e. 22 mm, according to NOAA Weather data)? I would assume that precipitation in July would be very intermittent in this region, and having -6 mm/day during one specific month of a specific year would not mean much;
- II. 257-258: this could be already stated at I. 146;
- II. 262-263: for sake of clarity, these boxes could be highlighted on Figure 5;
- I. 267: this should be compared with the available literature mentioned above;
- II. 318-319: the role of topography seems to be crucial here. Can the authors comment on this?
- II. 348-350: I find this argument a bit illogic. I see no reason why the sum of percentages from ATL and PAC sensitivity experiments would result in percentage increase of GLB23 minus the residual ("synergic interbasin interactions"). It might be informative to obtain composite mean maps for the three experiments, as in Figure 7, then subtract from the complete SST prescribed field to have an idea of the role of interbasin interactions. I do not think though, that the same can be done to infer the relative role of the three components for the overall GLB23 increase in extremes. The role of Atlantic and Pacific alone could determine a growth in the occurrence of the chosen events that would be larger than the combined effects, i.e. the interbasin interaction role would be effectively to reduce the role of the individual basins. This would not be captured if using the described approach, and it would be particularly relevant when dealing with extreme events. Please clarify this;
- I. 375: once again, it is not clear what you mean by "western North America", and I do not think that your arguments are equally valid for the whole of western North America, e.g. including Canada and Alaska;
- II. 412-419: this paragraph does not seem to belong to conclusions, as it would be better fitting into Introduction;
- II. 426-427: is there a specific reason why authors decided to use this Reanalysis product instead of others? What is the spatial resolution of this product?
- II. 451-452: the choice of 850hPa temperature seems a bit arbitrary, 2-metres temperature would have been a more valid alternative for near-surface temperature;
- II. 472-473: although I can understand that for the sake of comparison with Reanalyses, it was necessary to choose a reference period that was also encompassed by JRA55, it is a bit odd that a climatology for a period characterized by a strong positive trend and so close to the 2023 event has been considered. Perhaps a previous period (same length, still encompassing the 1958-present JRA55 coverage) would have been a more appropriate choice;
- I. 477: how is this ensemble created? What initial conditions are chosen, and how distant from each other? This should be made explicit;
- II. 480-481: same as in my previous comment. There are several bootstrapping techniques that could be used for the scope. For the sake of reproducibility, the technique shall be described here;
- II. 483-484: it is not clear if ATL23 and PAC23 are ensembles as well;

References

- Hebert et al. 2022: <https://www.nature.com/articles/s41561-022-01056-4>
- Jiang et al. 2022: <https://agupubs.onlinelibrary.wiley.com/doi/10.1029/2023GL103777>
- Kuhlbrodt et al. 2023: <https://journals.ametsoc.org/view/journals/bams/105/3/BAMS-D-23-0209.1.xml>
- Rex et al. 1950: <https://www.tandfonline.com/doi/abs/10.3402/tellusa.v2i4.8603>
- Richter et al. 2023: <https://agupubs.onlinelibrary.wiley.com/doi/full/10.1029/2024GL111563>
- Stoll et al. 2023: <https://wcd.copernicus.org/articles/4/1/2023/>

Reviewer #2

(Remarks to the Author)

Summary:

Overall, this is an interesting and well-written paper. Lopez et al. investigate the extreme and persistent heatwave that impacted the southwestern U.S. and northern Mexico in the summer of 2023. Using observational data, regression analysis, and atmospheric model simulations, the study finds that warm Atlantic SSTs, combined with a developing El Niño in the Pacific, significantly contributed to the increase in heatwave frequency, duration, and intensity. The authors employ heat budget analysis to identify key thermodynamic and radiative processes that sustained extreme surface temperatures. The study concludes that the interbasin synergy between the Atlantic and Pacific basins exacerbated the conditions that led to the record-breaking heatwave. The findings suggest that continued ocean warming may lead to an increased likelihood of prolonged and extreme heat events in the future. I do not have any substantial comments about the analysis or the main conclusions, which seem solid and well-supported. I recommend some minor revisions before this manuscript can be published, please see my comments below.

Main comments:

1. Some of the narrative and factual statements in the abstract and introduction can use finetuning, and I have left a number of specific line comments with my suggestions for changes.

2. The multitude of different time periods used for baselines seems unnecessary. For example, depending on the variable, the authors use 1979-2022 (SST and vertically integrated moisture transport), 1982-2023 (Phoenix temperatures), 1979-2023 (heat budget analysis and NAM), and 1960-2023 (SST regression). Could the authors find one or maybe at most two common baselines to use for all the analyses?

Specific comments:

Line 14: WMO 2023 reference not in citations.

Line 18: Hazard >> disaster

Line 18: Was this the costliest natural disaster of 2023 in North America? Perhaps specify this. Globally, I believe the Turkey earthquake in February was more costly.

Line 32-33: Is there a citation for "200 deaths"? I believe the death toll in Maricopa County alone was more than 200.

Line 35: Similar to my earlier comment, it should be specified that this was the costliest natural disaster in North America, not globally.

Line 36-37: I think the more relevant statistic to highlight is the record streak of days above 110F in Phoenix (31 straight), which broke the previous record of 18 straight days. Days >40°C may not be the most relevant statistic for Phoenix where the average high temperature in July is 41.4°C, meaning that 40°C is a cool day. The record warm Tmin of 97°F is interesting and can be kept in the text.

Line 38-40: This sentence, as currently worded, is inaccurate and should probably be deleted. The heat wave that produced the all-time record in New Orleans was a completely separate synoptic event that occurred on August 27 in a different region of the country. Similarly, the San Juan record resulted from a different tropical heatwave that was not part of the western North American heatwave that the authors are studying.

Line 60: delete "the"

Line 73: Might be good to add a quick sentence that explains how a Gill-type response can produce blocking over the Great Plains. I am assuming that the broad readership of Nature Communications will not be familiar with this.

Line 83-84: I would delete this sentence, since the AMO itself has been called into question as a "real" mode of internal climate variability.

Line 79-93: The effect of the 2020 international shipping sulfate reductions over the N. Atlantic might need to be mentioned in this paragraph, since they likely played a role in the warm SST during 2023.

Figure 1b: It might be good to adjust the x-axis to make the highest Tmax more visible. For example, Phoenix hit 122F (50C) in 1990 and 121F (49.4C) in 1995 but these temperatures are not visible in the histogram, making it seem like 119F (48.3C) in 2023 was the highest tmax between 1982-2023.

Line 113: This is incorrect, the 2023 maximum of 119F/48.3C was tied for the 3rd warmest maximum in Phoenix after 122F in 1990 and 121F in 1995 (see my comment above).

Line 117: "engulfed" >> "affected"

Line 124-125: Why highlight this particular Tmax on Aug 23 in New Orleans? They were even hotter on Aug 27th, when their all-time record high of 105F was reached. Perhaps this should be highlighted instead.

Figure 4a: It is surprising that 2020 doesn't appear as a very strong negative anomaly. This was a well-known failed monsoon season in the SW US, e.g.: Hoell et al., 2022: Record low North American monsoon rainfall in 2020 reignites drought over the American southwest. The authors may want to double check their monsoon index calculations, and if found to be correct, perhaps provide the appropriate caveats as to why their index does not capture certain anomalous years.

Line 197-201: Related to the above comment, the methodological details of the monsoon index calculations should be fully explained in the methods section.

Line 246: Please check the "(negative)", it doesn't seem to match the rest of the sentence.

Line 261: I am confused by "partial regression". Do the authors mean partial least squares regression?

Line 346: I may have missed it, but I don't recall seeing the extreme value analysis described in the methods?

Line 387: "persistent heat wave" >> "persistence of the heat wave"

Line 416-419: The statement about the earliest 110°F on record in June 2024 is only true for Las Vegas, not Phoenix.

Line 429: "Climate" >> "Centers"

Line 501: Delete "the"

Reviewer #3

(Remarks to the Author)

"The longest-lasting 2023 western North American heat wave was fueled by the record-warm Atlantic Ocean" by Lopez et al.

Summary:

This study investigates the causes of the longest-lasting heat wave in western North America in 2023 through observational analysis and atmospheric general circulation model (GCM) experiments. The key conclusion is that the 2023 heatwave was significantly driven by record-warm Atlantic sea surface temperature (SST) and growing El Niño in the Pacific. The authors conducted a comprehensive analysis of the roles of both the atmosphere and the ocean in driving the heat wave, using energy budget analysis, 3-dimensional atmospheric circulation and moisture transport, and the impact of SST over both the Atlantic and Pacific Oceans. I believe this manuscript is well-written and makes a valuable scientific contribution to understanding the dynamic and thermodynamic changes that triggered this heatwave event. However, I found several issues that need to be addressed to improve the manuscript. For publication, please revise the manuscript to satisfactorily address the specific comments provided below.

Specific Comments:

1. The authors concluded that the exceptionally warm Atlantic Ocean and the growing El Niño in the Pacific caused changes in circulation, leading to an anticyclonic blocking pattern during the summer of 2023. They supported their findings with evidence of interbasin synergy effect captured in model experiments. However, it remains unclear about how the Pacific SST interacts with the Atlantic in this synergy effect to create the persistent blocking pattern. The dynamic mechanism needs to be addressed as to how the El Niño-driven SST in the Pacific is linked to the ongoing atmospheric blocking over the western U.S.
2. Potential driver of the Atlantic warming in the discussion section: I'm still not convinced about the possible impact of El Niño on the Atlantic SST increase (e.g., L404-406), as warmer-than-average Atlantic SST was not frequently observed during the previous El Niño developing years. Could you provide a more compelling explanation of the El Niño-driven increase in Atlantic SST?
3. This study shows many results for Phoenix, Arizona (e.g., Figures 1, 3b,c, and 9). Extremely high temperatures were, however, observed in other regions as well, including Texas, northern Mexico, and Louisiana. I recommend that the authors provide evidence showing that these other areas share quite similar characteristics to Phoenix, or at least discuss these regions in greater detail.

4. Equation 4 for budget analysis: I believe vertical integration over a range of pressure levels would be more preferable than considering a single level (850hPa). This would be specifically true for vertical advection.
5. Advection in budget analysis: Could you elaborate which factor, the wind versus temperature gradient, and mean versus anomaly, has a greater influence on determining the advection in the budget analysis?
6. The authors mention that they present the PV pattern in Figure 3 during the peak of the heatwave. However, it seems this is based on the heatwave that occurred in Arizona. I am wondering how the large-scale atmospheric condition shown in Figure 3 could support the development of heatwave condition over E Mexico, E Texas and Louisiana, where northerly winds are prevalent. Is the PV pattern here explaining the large-scale condition for the heatwave over Arizona only?
7. Following the comment above (#6), the timing of the heatwave peak seems to vary across different regions, such as Arizona, Texas, Louisiana, NW Mexico, and E Mexico, as seen in Supplementary Figure 1. The PV and circulation pattern during the peak (June, July, or August) of the heatwave could differ between regions. The authors may need to address this issue.
8. Anomalous moisture transport: Figure caption 4 states this field is computed from departures from climatology. Does this mean that the departure of wind and moisture from climatology is used to compute the anomalous moisture transport and its convergence? If so, I believe this approach may be incorrect. Moisture transport is determined by four components: 1) advection of mean moisture by mean wind, 2) advection of mean moisture by anomalous wind, 3) advection of anomalous moisture by mean wind, and 4) advection of anomalous moisture by anomalous wind. The fourth term may be an order of magnitude smaller than the other terms. I recommend that authors clarify how the Figure 4c pattern was calculated.
9. The moisture transport field suggests that the heat transport from the Atlantic might also be lower than climatology, potentially due to the prevailing wind direction, which is not southerly. Could you offer some discussion on this?
10. L249: Is it "significantly" negative? The negative anomaly observed in June weakens in July. Additionally, could you explain how this negative anomaly in the Atlantic is related to the anticyclonic circulation linked to atmospheric blocking in the upper troposphere? Are they connected, or is the blocking a consequence of upper-level dynamics with no influence from the surface level shown in Figure 5?
11. The authors present Figure 5 to explore large-scale climate factors that may drive heatwave conditions. However, in terms of causality, it is not clear whether the precipitation over U.S. land is a cause or an effect of the heatwave. This needs to be explicitly addressed in the manuscript.
12. Is the upper-level anticyclonic circulation pattern in Figure 6a (contoured) similar to the observed pattern in 2023? If so, is this pattern generally pre-dominant throughout the summer?
13. Why does the Atlantic-only experiment produce warmer than average temperature along the western coast of Mexico, with no notable warming over the central to eastern side of Mexico (Figure 7)?
14. While the amplitude change of heatwave events in the AGCM experiments shows the highest values over Mexico, the number of events, heatwave days, and duration in Figure 8 show the greatest increase in the far inland areas like Colorado, as well as western Mexico and the southern coastal region. Have you considered why the largest values appear in the far inland area? Could the large-scale atmospheric pattern in the AGCM experiment offer any insights to help explain this?

Version 1:

Reviewer comments:

Reviewer #1

(Remarks to the Author)

I acknowledge that the manuscript

"The longest-lasting 2023 western North American heat wave was fueled by the record-warm Atlantic Ocean" by Hosmay Lopez, Sang-Ki Lee, Robert West, Dongmin Kim, Liwei Jia

featured substantial improvement compared to the previous version. Particularly, I appreciated the great deal of additional work that the authors put into the manuscript, thoroughly addressing comments by myself and the other reviewers (and also proving that I was incorrect about a few remarks!). I do acknowledge that the amount of information provided is now fairly larger and the message is more complete and robust, also involving aspects of the large-scale circulation that were barely touched upon in the previous manuscript. I do have a few minor remaining comments that I want to raise, mainly regarding additional references and better context on a few aspects.

Provided this, I do believe that the manuscript would be ready for publication.

MINOR COMMENTS

ll. 29-30: given that the authors acknowledge the spatial compounding nature of the event, it would be nice if the authors could mention already here something about the relevance of what they find for the large-scale atmospheric circulation, especially given that they adopt the strategy of looking into inter-basin synergies and the role of planetary-scale Rossby waves (e.g. Lembo et al. 2024).

Lembo, V., and Coauthors, 2024: Dynamics, Statistics, and Predictability of Rossby Waves, Heat Waves, and Spatially Compounding Extreme Events. *Bull. Amer. Meteor. Soc.*, 105, E2283–E2293

ll. 59-61: quasi-resonant amplification is not the only possible explanation for large-scale atmospheric circulation signature on heatwaves (see reference above and literature cited therein);

ll.73-74: I think it is very relevant to mention here a recent paper (England et al. 2025) on the drivers of the North Atlantic SST exceptional anomalies;

England, M.H., Li, Z., Huguenin, M.F. et al. Drivers of the extreme North Atlantic marine heatwave during 2023. *Nature* (2025). <https://doi.org/10.1038/s41586-025-08903-5>

ll. 107-110: In this respect, it is also worth mentioning that the tropical atmospheric bridge here mentioned is linked itself with the extratropical Atlantic variability, as argued for instance in An et al. 2021;

An, X., B. Wu, T. Zhou, and B. Liu, 2021: Atlantic Multidecadal Oscillation Drives Interdecadal Pacific Variability via Tropical Atmospheric Bridge. *J. Climate*, 34, 5543–5553

Figure 2: I still think that Figure R2 is actually more informative than the boxes approach (for instance, showing the relative magnitude of anomalies, and some peculiar behavior, such as for instance the one exhibited by latent heat fluxes), but I acknowledge the readability issue and therefore leave it to the authors;

Reviewer #2

(Remarks to the Author)

I thank the authors for addressing my points. I believe this manuscript is ready to be published pending the approval of the other reviewers.

Reviewer #3

(Remarks to the Author)

I appreciate the authors for revising their manuscript. My previous concerns have been addressed satisfactorily, and I have no remaining major issues. I recommend accepting the revised version as it stands.

made.

Please extend our gratitude to the three anonymous reviewers for their constructive comments/suggestions that have led to a significant improvement to our manuscript.

The following overarching changes have been made to the manuscript in order to address the reviewers' comments/suggestions:

- Added more up-to-date statistics and chronological description of the heat wave, including the impact of the heat wave in several cities besides Phoenix, Arizona.
- More in-depth analysis on the physical mechanisms driving the heat wave, including clarifying dynamic mechanisms and equations
- Added more context in relation to existing literature

In what follows, we respond to each comment of all reviewers. Our responses are in regular font and the original reviewers' comments are in blue italic font. Each comment is addressed in the revised manuscript. We point to specific line numbers where the changes were made, these line numbers refer to the lines of the tracked-changes version of the text, A copy of the revised manuscript is provided with tracked changes revisions.

Reviewer #1 (Remarks to the Author):

The manuscript

“The longest-lasting 2023 western North American heat wave was fueled by the record-warm Atlantic Ocean” by Hosmay Lopez, Sang-Ki Lee, Robert West, Dongmin Kim, Liwei Jia

features an in-depth investigation of the causes and processes related to the outstanding heatwave that affected southwestern US and Mexico between June and July 2023. By analysing in-situ measurements, remote sensing, gridded observations and reanalyses, the authors argue that the event in the region of interest, and particularly in the area of Phoenix, was characterized by abnormal positive anomaly of sensible, diffusive, and radiative surface fluxes towards the surface. They also found that an anticyclonic pattern, suppressing advective fluxes and enhancing local adiabatic heating, were responsible for the long heat streak across much of July. The North American Monsoon, and moisture fluxes entering the area from the oceans, were anomalously weak. Regression analyses evidenced that the role of North Atlantic SSTs was more relevant than that associated with subtropical Pacific SSTs. This was confirmed by sensitivity analyses using a General Circulation Model in AMIP settings, forcing the atmosphere with prescribed SSTs representative either of the overall SST anomalies in Summer 2023, or of the anomalies over the Atlantic and over the Pacific during the same period. The interbasin feedback is claimed to play a substantial role in the build-up of the anomaly, therefore the authors stress that the action of both basins has to be taken into account when attempting to attribute the North American heatwaves to anthropogenic climate change enhanced by oceanic natural variability.

Overall, I think that the manuscript has scientific merits, and is worth publication. However, there are several substantial aspects that should be carefully taken into account before the manuscript can be accepted.

My main concern is that in the section involving observations authors fall a bit short in justifying the correlations of different aspects of the circulation with process understanding. Another aspect that should be clarified is the attribution of extreme events to the role of Atlantic-Pacific-synergy of both SSTs. Therefore, I recommend the manuscript for major revision.

Response: We are very grateful for your constructive comments/suggestions and hope to have successfully addressed all your comments. Please find our replies to your specific comments below in regular fonts and your original comments in blue italic fonts. We point to specific line numbers where the changes were made, these line numbers refer to the lines on the tracked-changes version of the paper.

Specific comments are detailed below.

- l. 27: 2024 surpassed 2023 as warmest year on record. Please mention it.

Response: Agree. We did not add this fact in the original version as the final report for 2024 was not released by the time we submitted the paper for peer-review (November 2024). The discussion section now talks about 2024 extreme temperatures and we also have acknowledged this fact in the introduction of the revised manuscript as follows (Lines 28 – 30):

“The year 2023 was the second warmest on records^{1,2}, only surpassed by 2024 (Monthly Global Climate Report for Annual 2024). Of particular notoriety was the 2023 boreal summer months (i.e., June-August), which were characterized by more than 20% of the land surface setting new extreme warm records and multiple heatwave events over most continents³.”

NOAA National Centers for Environmental Information, Monthly Global Climate Report for Annual 2024, published online January 2025, <https://www.ncei.noaa.gov/access/monitoring/monthly-report/global/202413>. DOI: <https://www.ncei.noaa.gov/access/metadata/landing-page/bin/iso?id=gov.noaa.ncdc:C00672>

- l. 36: Is there any reason why Phoenix would be an exceptional spot where to focus the analysis? The rational of the work would be certainly improved by providing a justification for this;

Response: Agree. We used Phoenix as an example of a major metropolitan area impacted. In addition, Phoenix also provides one of the longest reliable and continuous observational records in the area of interest. With that said, the focus of our paper is on the heat wave event as a whole, which was fairly large-scale and exerted severe impacts throughout the southwestern U.S. and Mexico.

We have now added more discussion on the effects of the 2023 heat wave on other important population centers. For example, we added a new supplementary figure (Supplementary Fig. 1) and discussion (Lines 142 – 144) in the revised paper showing the temperature evolution during the summer of 2023 for Las Vegas, Nevada, Albuquerque, New Mexico, El Paso, Texas, and San Antonio, Texas. Note that all these cities were impacted by the heat wave with both maximum and minimum temperatures exceeding their corresponding 95 percentile for multiple days. Also note that the analysis only includes cities within the continental U.S. as long-term continuous and reliable data are not available for northern Mexico.

- ll. 43-45: *I see that all these examples mention Northern Hemisphere continents, but outstanding heatwaves also affected the Southern Hemisphere. Perhaps worth mentioning it.*

Response: Agree. We have added examples of Southern Hemisphere heat waves, which 2023 was an exceptional year for global impacts. For example, South America experienced major heat waves, with record-breaking temperatures (Perkins-Kirkpatrick et al. 2023). Heat waves are also the leading cause of natural hazards related deaths in Australia, accounting for more than 55% of the reported fatalities (Coates et al. 2014). See Lines 54 - 58 of the revised text.

Coates, L., Haynes, K., O'Brien, J., McAneney, J., & De Oliveira, F. D. (2014). Exploring 167 years of vulnerability: An examination of extreme heat events in Australia 1844–2010. *Environmental science & policy*, 42, 33-44.

Perkins-Kirkpatrick, S., Barriopedro, D., Jha, R., Wang, L., Mondal, A., Libonati, R., & Kornhuber, K. (2024). Extreme terrestrial heat in 2023. *Nature Reviews Earth & Environment*, 5(4), 244-246

- ll. 53-54: *it might be worth mentioning that the effects of heatwaves are also exacerbated by increasing urbanization, agricultural loss, increased aridification;*

Response: Great point, we added the role of increased urbanization and land use changes and the following references to the revised text (Lines 64 - 65).

Fischer, E. M., Oleson, K. W., & Lawrence, D. M. (2012). Contrasting urban and rural heat stress responses to climate change. *Geophysical research letters*, 39(3).

Mishra, V., Ganguly, A. R., Nijssen, B., & Lettenmaier, D. P. (2015). Changes in observed climate extremes in global urban areas. *Environmental Research Letters*, 10(2), 024005.

Overpeck, J. T., & Udall, B. (2020). Climate change and the aridification of North America. *Proceedings of the national academy of sciences*, 117(22), 11856-11858.

- l. 75: *furnished -> provided*

Response: Done.

- l. 77: *I may agree that the oceanic timescales have timescales that are longer than those pertaining on local heat extremes, but I am not sure this applies to land process variabilities as well (e.g. Hebert et al. 2022);*

Response: The intention of this sentence was to onto potential oceanic (mostly SST) predictors for heat waves at the seasonal timescales. This stems from the fact that, at seasonal and longer timescales, the predicted portion of a climate variable comes mostly from oceanic processes (modes of variability) (Fig. 2 in Mariotti et al. 2018). For example, seasonal outlooks like temperature and precipitation over the U.S., basin wide tropical cyclone activity, etc. draw their predictors primarily from the global SST state.

The study of Hebert et al. 2022 reports on the longer term (years to centuries) variability over regional terrestrial domains and how it may be underrepresented in future projections from climate models,

especially for regions with significant maritime influences. That is, these regions would experience even larger natural variability than what is currently not well represented in GCM projections, which is shown to be more relevant at timescales longer than multi-decadal. Their finding has great implications as this enhanced natural variability would cause unpredictable climate shifts as opposed to a more homogeneous projected warming.

However, this is somewhat unrelated to our argument stated in the sentence as we are not referring to future projected warming versus natural factors, instead we are interested in the predictable aspects of natural variability at the interannual timescales, which mostly comes from ocean variations. With that said, we have added a brief discussion and the reference to Hebert et al. 2022 in the revised text (Lines 93 - 95).

Mariotti, A., Ruti, P. M., & Rixen, M. (2018). Progress in subseasonal to seasonal prediction through a joint weather and climate community effort. *Npj Climate and Atmospheric Science*, 1(1), 4.

- II. 82-83: *Perhaps it is worth mentioning Kuhlbrodt et al. 2023 on the role of ocean heat content for North Atlantic 2023 extreme;*

Response: We added the reference Kuhlbrodt et al. 2023 to the revised text (Line 97).

- I. 97: *it is surprising that the authors seem to be unaware of the active debate on the interbasin Atlantic-Pacific teleconnections, especially in the Tropics, that has developed in years. I am not in the position to provide a conclusive literature survey, but I may suggest that the authors have a look at the debate around the Jiang et al. 2023 paper (e.g. Richter et al. 2024) and provide some references in the manuscript.*

Response: We are aware of the current debate regarding the Atlantic-Pacific Interbasin interactions. Some of the co-authors here in our paper have recent publications regarding the pantropical interactions between the three (Indian, Pacific, and Atlantic) basins (Lee et al. 2023) and how the Pacific-Atlantic interannual variability could explain extreme events like major hurricanes (Kim et al. 2023).

The papers of Jiang et al. 2023 and references therein and the comments from Richter et al. 2024 are centered around the equatorial Pacific-Atlantic interactions, mostly with regards to how El Niño-Southern Oscillation (ENSO) influences and is influenced by its Atlantic “little cousin”, Atlantic Niño. However, our paper instead focuses on the impact of tropical Atlantic and Pacific warm sea surface temperatures on North American heat waves. Thus, the physical mechanisms driving these SSTs anomalies, including the tropical Pacific/Atlantic interaction conundrum, are beyond the scope of our work. We made this distinction in the revised paper and added Jiang et al. 2023 and Richter et al. 2024 (Lines 121 - 124).

Lee, S. K., Lopez, H., Tuchen, F. P., Kim, D., Foltz, G. R., & Wittenberg, A. T. (2023). On the genesis of the 2021 Atlantic Niño. *Geophysical Research Letters*, 50(16), e2023GL104452.

Kim, D., Lee, S. K., Lopez, H., Foltz, G. R., Wen, C., West, R., & Dunion, J. (2023). Increase in Cape Verde hurricanes during Atlantic Niño. *Nature Communications*, 14(1), 3704.

- l. 101: related to my previous comment, it could be better emphasized that the findings that are actually contributing substantially to the topic regard the inter-basin synergy. So far, it seems to the reader that is just one of the many aspects touched here in the context of the 2023 heatwave;

Response: As per our previous reply, our focus is not on explaining the tropical dynamics of SST variations. Instead, our focus in this paper is with regards to remote impacts. What we refer to as the “Interbasin synergy effect of Pacific-Atlantic forming” is in reference to the role of SSTs on heat waves. We politely disagree with the reviewer in that we believe that performing a literature review on ENSO-Atlantic Niño interactions will cause some distraction and steer the focus of this paper away from the 2023 heat wave event.

With that said, we agree that more clarification needs to be made with regard to the “interbasin synergy effect”. For that, we added a statement clarifying that this interbasin synergy is referring to the constructive interaction between the warm Atlantic and ENSO in modulating the heat wave occurrence rather than alluding to the tropical Pacific/Atlantic interaction conundrum (i.e., Jiang et al. 2023; Richter et al. 2024). See Lines 121 - 124 of the revised text.

- l. 104: it shall be mentioned explicitly here what is the reference period for the computation of the climatology;

Response: The reference period for the computation of climatology is the 1979 – 2022 period as this is the overlapping period for all observational and reanalysis datasets used in the paper. We have clarified this in the revised text’s Method section.

- l. 109: I struggle a bit to read Figure 1. I wonder why the authors prefer showing absolute temperatures instead of anomalies, that would much better emphasize the T95 exceedances;

Response: One of the reasons to show the absolute temperature this way is that it is more efficient when showing both minimum and maximum temperatures in fewer numbers of figure panels. If we choose to show anomalies instead, then separate panels for a) and b) will be needed for minimum and maximum temperatures as these will need to be centered by a zero mean.

However, we agree that some of the details are difficult to depict. Therefore, we have modified Fig. 1 to more easily depict the exceedance over the T95. For example, panel a) shows the mean temperature (thick-dashed line), T5 to T95 range (filled color range), and the actual absolute temperature thick line. The exceedances over the T95 are now highlighted by the red shading which now includes red-dot hatching, signifying when the absolute temperature overshoots the T95 threshold. See Fig. R1 below which is now the new version of Fig. 1.

Figure R1. Heat wave event of 2023. A) Seasonal evolution of maximum (red) and minimum (blue) temperature for Phoenix, Arizona for the year 2023 from 1 June to 20 September. The long-term daily mean is shown by the dashed line whereas the 5th and 95th percentiles are shown by the shading region. Excess above the 95th percentile is shown by red shading plus hatching for both maximum and minimum temperatures. B) Observed histogram of maximum (red) and minimum (blue) temperature for Phoenix, Arizona for the period 1 June to 31 August from 1955-2023. The extremely warm temperatures of 19 July 2023, during the peak amplitude of the heat wave, are shown for reference. c) Five-day averaged surface temperature anomaly centered on 19 July 2023.

- l. 129: Figure 2 is very chaotic in my opinion, and I think it shall be improved by showing actual time series, instead of boxes with 10-days averages. If the daily spread is too high, a 10-days moving average could be superimposed on the actual time series. Also, I think that it is very misleading to position color bars next to x-axes, whose labels are only found at the bottom of the figure;

Response: We politely disagree. The reason such an approach was chosen is twofold. (1) There is a lot of information to convey in Fig. 2 (14 timeseries in total), so interpretation of the signal and the relative strength of each entry is difficult, requiring multi-panels due to different units (see Fig. R2 below). (2) The second issue is that given the large number of timeseries shown, using a color-blind neutral scheme for each of the 14 lines is extremely difficult, as some of the co-authors themselves are color blind. The original approach of boxes sharing the same color palette and color bar (blue \rightarrow red) among all variables is a fix to these two issues. It is also easier to interpret a very busy figure at a glance, given the homogeneity in color scheme and signal amplitude.

Also note that due to another reviewer's suggestion, we added an additional entry for the 2-meter air temperature anomaly. Please note that we also added x-axis labels to all sub-panels for easier representation of the time axis. See revised Fig. 2.

Figure R2. Heat budget analysis. Energy budget averaged every 10 days over the southwest U.S. and northwest Mexico from 6 June to 24 August 2023. A) 200 hPa geopotential height anomaly [gpm], 850 hPa temperature and 2-meter temperature anomalies [°C]. b) Vertically integrated anomalous heating rates from 975 – 800 hPa layer [°C day⁻¹], and c) surface heat fluxes (net surface shortwave and longwave radiation, sensible and latent heat fluxes) [Wm⁻²]. Daily anomalies are derived from the long-term monthly mean for the 1979-2023 period.

- I. 140: Figure 3c may be obvious to a reader with some sort of meteorology background, but it is not at all for many readers of Nature Communications. Therefore, the skew-T should be carefully explained

(perhaps in the Methods?) making a bit more explicit what the authors discuss, particularly at lines 184-188 and in Figure 3c description;

Response: Agree. We added a more detailed explanation of the skew-T plot and what each line represents for the untrained reader (Lines 220 - 221). Also note that this reference to the skew-T plot and Fig. 3 in general was premature, so we deleted the clause and introduced Fig. 3 in more detail later in the revised paper, which should aid the reader in interpreting the diagram.

- l. 142: it is nowhere explained what the authors mean by “diffusive heating” and how this is computed;

Response: Diffusive heating rate is a variable provided directly by the Japanese reanalysis JRA55, which entails vertical diffusive heating (K / day). This is a motivation for using the JRA55 analysis as it provides various diabatic heating rate fields at pressure levels, which other reanalysis products do not offer.

The total diabatic heating rate (i.e., the Q_{net} term in the thermodynamic energy equation eq. 4) consists of the net atmospheric radiation (short and longwave), vertical diffusion heating rate, and latent heat from moist processes (including the large-scale condensation heating rate and convective heating rate). The vertical diffusive heating (i.e., sensible heating from the surface up into the air column) and adiabatic heating (due to subsidence) have been shown to play dominant roles in exacerbating surface warming during heat waves (Zhou et al. 2024). The JRA55 heating rates have been extensively used in other studies of atmospheric dynamics (Ma and Franzke 2021; Shi et al. 2020; Lopez et al. 2022).

We added this explanation to the Method section (Lines 496 - 500). However, more details on the definition of the heating rates can be found at: <https://www.jma.go.jp/jma/jma-eng/jma-center/nwp/outline2013-nwp/index.htm>

Lopez, Hosmay, Dongmin Kim, Robert West, and Ben Kirtman. “Modulation of North American heat waves by the tropical Atlantic warm pool.” *Journal of Geophysical Research: Atmospheres* 127, no. 21 (2022): e2022JD037705.

Ma, Qiyun, and Christian LE Franzke. “The role of transient eddies and diabatic heating in the maintenance of European heat waves: a nonlinear quasi-stationary wave perspective.” *Climate Dynamics* 56 (2021): 2983-3002.

Shi, Ning, SuolangTajie, Pinyu Tian, Yicheng Wang, and Xiaoqiong Wang. “Contrasting relationship between wintertime blocking highs over Europe–Siberia and temperature anomalies in the Yangtze River basin.” *Monthly Weather Review* 148, no. 7 (2020): 2953-2970.

Zhou, Jie, Haipeng Yu, Peiqiang Xu, Wen Zhao, Bofei Zhang, Yu Ren, Shanling Cheng, Yongqi Gong, and Yaoxian Yang. “Extreme heat event over Northwest China driven by Silk Road Pattern teleconnection and its possible mechanism.” *Atmospheric Research* 297 (2024): 107090.

- l. 146: similarly to my comments above: it is not worth mentioning that the Bowen ratio increases if it is not explained what this means in terms of process understanding;

Response: Agree.

The Bowen ratio is used to link water and energy balances of the climate system. A Bowen ratio increase, often present during mega heat waves (Miralles et al. 2014), suggests a reduction in the evaporation/evapotranspiration and an increase in the sensible heating and thus increase in near surface temperatures. We added more explanation and reference to the revised text (Lines 175 - 177).

Miralles, D. G., Teuling, A. J., Van Heerwaarden, C. C., & Vilà-Guerau de Arellano, J. (2014). Mega-heatwave temperatures due to combined soil desiccation and atmospheric heat accumulation. *Nature geoscience*, 7(5), 345-349.

- l. 159: *wouldn't it be clearer to state that the dynamical tropopause crosses the 2 PVU isosurface where the thick solid line is?*

Response: The dynamical tropopause is defined here as the 2 PVU isosurface. We clarified this in the revised text (Line 193).

- l. 165: *not clear why I would infer from Figure 2 that there is shortwave surface heating in clear sky, i.e. why I would have to assume clear sky;*

Response: Agree. While the skew-T diagram (Fig. 3c) shows a very dry vertical profile, i.e., a well-separated air temperature (red) and dew point temperature (blue), it is not obvious from Fig. 2 that there are clear sky conditions. To show this, we have computed the cloud radiative effect from the reanalysis during the middle of the heat wave (Fig. R3 below). This effect is easily extracted by taking the difference between the clear-sky minus the all-sky radiative fluxes at the top of the atmosphere (TOA) following Kobayashi et al. 2015. Note the fairly large swath area of no clouds over the southwestern U.S. (our study region), which is evident by the near-zero net cloud radiative effects, including shortwave (Fig. R3b) and longwave (Fig. R3c) effects. We also obtained a similar pattern using direct retrieval of radiative effects from the Cloud and the Earth's Radiative Energy System (CERES) observations.

We added Fig. R3 to the revised manuscript as Supplementary Fig. 4.

Kobayashi, S., Ota, Y., Harada, Y., Ebata, A., Moriya, M., Onoda, H., ... & Takahashi, K. (2015). The JRA-55 reanalysis: General specifications and basic characteristics. *Journal of the Meteorological Society of Japan. Ser. II*, 93(1), 5-48.

Figure R3. Cloud radiative effects computed from the differences between clear-sky minus all-sky radiative fluxes at the top of the atmosphere (TOA) for a) net fluxes, b) shortwave reflected fluxes, and c) outgoing longwave fluxes. These values represent the five-day average centered on 19 July 2023, during the middle of the heat wave. Regions with dark blue represent little or no clouds.

- l. 168: it is textbook literature that atmospheric blocking is among the most prominent features of extratropical low frequency variability (e.g Rex et al. 1950), then some sort of agreement shall be reached on what the authors refer to as “synoptic”, because blocking is not proper of synoptic timescales (although it might be to some extent of synoptic spatial scales, although I would rather refer to it as a manifestation of planetary waves; cfr. Stoll et al. 2023 and references therein)

Response: Agree. Atmospheric blocking events, such as “omega”, “high-over-low”, and “Rex” blocks, are often semi-persistent due to their underlying dynamical and thermodynamical forcing. Several of these blocks have been observed to persist for weeks (e.g., the Russian heatwave and the event studied in

this work). Thus, we changed the language to reflect that these are manifestations of planetary waves and added Stoll et al. 2023 to the reference (Line 203).

- l. 198: What is the exact definition of “western North America” here?

Response: Here we geographically define western North America as the region enclosed by 15°N-50°N and 125°W-90°W. However, with regards to the exact boundaries of what constitutes the North American Monsoon (NAM), we follow the concept of the global monsoon (Wang and Ding 2008; Lee and Wang 2014). This index is defined as where the local summer (May – September) precipitation minus the winter (November–March) precipitation exceeds 2.5 mm per day and the local summer precipitation exceeds 55% of total annual precipitation (Lee and Wang 2014). We applied these constraints over the geographical box defined above, which is shown in Fig. R4, depicting the location of the NAM index by the red stipples, which is the location meeting the NAM definition constraints.

We have clarified this in the revised text and added the red stipples to the monsoon associated precipitation plot (Fig. 4) and the definition of the NAM index in the Method section (Lines 546 - 550).

Wang, B., Ding, Q., & Liu, J. (2011). Concept of global monsoon. In *The global monsoon system: research and forecast* (pp. 3-14).

Lee, J. Y., & Wang, B. (2014). Future change of global monsoon in the CMIP5. *Climate Dynamics*, 42, 101-119.

Figure R4. June – September (JJAS) precipitation anomaly during 2023. The red stipples indicate the North American monsoon region defined as where the local summer (May – September) precipitation minus the winter (November–March) precipitation exceeds 2.5 mm per day and the local summer precipitation exceeds 55% of total annual precipitation. Dataset from the JRA55 reanalysis.

- l. 202: this is another example of one of my main concerns about (this section of) the manuscript. Besides showing that the North American Monsoon is weak in 2023, claiming that some sort of correlation is present between moisture fluxes enter the region and negative latent heat fluxes locally (in

Phoenix, at least), what do we learn about the relation between exceptional heatwaves in the region and the monsoon? Could it be just a chance? What happened in 1982 (the weakest NAM on records, according to the time series in Figure 4a)? Is the anomaly significant wrt. NAM climatology?

Response: We agree that more quantification of the relationship between the NAM precipitation and surface temperatures needs to be shown. For this we computed the NAM index (Fig. 5a) correlation with maximum July temperatures over the study region for the 1979-2022 period (see Fig. R5b below). Note that while this correlation does not include the year 2023 (done to prevent aliasing the relationship with 2023), the spatial pattern is very similar to the 2023 temperature anomalies over the southwest U.S. and Mexico (Fig. R5a) which shows warmer temperatures when the NAM is negative (note that the correlation colorbar in Fig. R5b was reversed intentionally to depict the negative correlations and for easier comparison between panels).

With regards to the year 1982, while this was the weakest NAM based on our definition, the temperature anomalies over the study region were just slightly above climatology (Fig. R5c). So, the year 1982 was an outlier in the NAM – maximum temperature relationship. In contrast, the year 2023 is much closer to the regression line (Fig. R5c) and thus fits the NAM – maximum temperature negative relationship.

We this analysis to the revised manuscript (Lines 238 - 241) and added Fig. R5 as supplementary Fig. 5.

Figure R5. A) July 2023 maximum 2-meter temperature anomalies. B) Correlation of maximum 2-meter temperatures and the North American Monsoon (NAM) index for the 1979 – 2022 period (the year 2023

was excluded from the correlation analysis intentionally to avoid aliasing the results). Note that the colorbar is reversed for easier comparison and reflection of the negative correlations. C) Scatter plot of the NAM index versus 2-meter air temperatures averaged over the southwestern U.S. and northern Mexico (black box in panel b). The best-linear fit is shown by the dashed line along with the years 1982 (blue) and 2023 (red). The correlation between the NAM and index and 2-m temperatures is $r = -0.58$, p -value < 0.01 . See Methods for definition on the NAM index.

Is the anomaly significant wrt. NAM climatology?

Response: The anomalous NAM index has a mean = 0.00 mm/day and an interannual standard deviation $\sigma = 0.61$ mm/day. Using a student-T test, with sample size $N = 45$ (years), the confidence interval is ± 0.25 at a 99% significance level from a student-T test. Thus, any NAM anomalies larger than 0.25 are significantly away from zero. We added the significance test (see Fig. R6) to the revised manuscript (Fig. 4a).

Figure R6. Time series of the North American monsoon index for June-August. Years of significant NAM index anomalies are highlighted by color-filled bars. Statistical significance is based on the 99-percentile significance level based on a Student’s-t test.

- l. 211: related to my previous comment; authors are comparing the NAM activity along the whole season, but the heatwave peaked during 14 days in July and short heatwave periods emerged during August and September (possibly not related to the major one). It would perhaps be more informative if the evolution of daily moisture transports could be tracked during the season;

Response: Please note that the main scope of this paper is not to describe the synoptic evolution of the heat wave, rather than investigating why the extreme heat was semi-persistent throughout the summer. That is, we are interested in the longer-term aspects of the event in association with forcing from large-scale interannual variability (e.g., ENSO, Atlantic SSTs, N. American monsoon). The evolution of daily moisture transports would be extremely noisy and driven by high-frequency weather patterns outside of the paper’s main scope. With that said, we agree that it would be informative to discuss the daily evolution of the NAM, which nicely depicts the weak rainfall and late onset in 2023 relative to climatology (Fig. R7).

The daily evolution of precipitation is divided into regional indices (a northern and a southern index), following Duan et al. 2024, as these are very distinct in their rainfall behavior. As shown in Fig. R7, the precipitation stays below climatology until the middle of August 2023 for the northern region for most of the summer. For the region over Mexico, the precipitation does not reach its climatological value until 9 July, well past the historical onset which occurs around 29 June. However, it then decreases well below climatology for most of the remaining active monsoon season.

Duan, S., Ullrich, P., & Boos, W. R. (2024). Meteorological drivers of North American monsoon extreme precipitation events. *Journal of Geophysical Research: Atmospheres*, 129(13), e2023JD040535.

Figure R7. Daily evolution of the North American Monsoon (NAM) indices for a) northern domain (blue region in the inserted map) and b) southern domain (red region in the inserted map).

- I. 254: how relevant is the negative anomaly (shown in mm/day) compared to the precipitation that usually occurs during July in the region (i.e. 22 mm, according to NOAA Weather data)?

Response: Very relevant and significant above the 95-percentile confidence level based on a Student's t-test (see Fig. R8 below). For example, the mean climatological July precipitation in the southwestern U.S. and northern Mexico peaks at around 10 mm per day (Fig. R8b). Thus, anomalies of around 5 mm per day (Fig. R8d) are fairly large in comparison. Please note that the units of precipitation used in our work are mm per day, so this is a precipitation rate instead of an aggregated/integrated rainfall amount, which we believe it is what the reviewer is alluding to. Also note that these precipitation rates are taken from monthly mean data. In addition, Fig. R8 below is a zoomed in version from Fig. 5e and f of the paper

With that said, we added significant tests to the analysis of Fig. 5 in the revised text and explicitly clarified that the rainfall units here are rainfall rates rather than total precipitation.

I would assume that precipitation in July would be very intermittent in this region, and having -6 mm/day during one specific month of a specific year would not mean much;

Response: We politely disagree with the reviewer. A -6 mm/day monthly mean precipitation rate anomaly is fairly significant for this region (Fig. R8), which was also shown to be the second driest NAM index in the 1979 – 2023 period (Fig. 5). Please note that this is a region with monthly mean climatological rainfall rates of around 10 mm/day. Even regions of significant rainfall amounts (Indian Monsoon region, Bay of Bengal, and tropical Pacific Ocean) only experience around 25 mm/day monthly mean precipitation rates in July (see: <https://gpm.nasa.gov/data/IMERG/precipitation-climatology#monthlyprecipitationclimatology>)

Also note that we are assessing monthly mean rainfall statistics and not daily precipitation and that a standard unit convention for rainfall rate is mm/day.

Figure R8. a and b) June and July precipitation rate climatology respectively. Panels c) and d) show the precipitation rate anomalies for June and July 2023 respectively. Black stippling in panels c and d) indicate statistical significance at a 95% confidence level based on a Student's t-test. The precipitation rate is measured in mm per day.

- ll. 257-258: this could be already stated at l. 146;

Response: Agree, we removed this clause as it is already mentioned.

- ll. 262-263: *for sake of clarity, these boxes could be highlighted on Figure 5;*

Response: Done

- l. 267: *this should be compared with the available literature mentioned above;*

Response: This is the first time, to the best of our knowledge, that the combined effect of Atlantic and Pacific SST anomalies on atmospheric circulation is investigated. However, we added references to the revised text describing their individual roles in generating these upper-level flow anomalies (Lines 324 – 327). For example, an El Niño event produces a tropospheric warming throughout the tropics via a fast equatorial Kelvin wave and off-equatorial anticyclonic anomalies, similar to Fig. 6e (Trenberth et al. 1998, Lee et al. 2018; Lopez et al. 2019). However, ENSO response to the northern hemisphere is strongest later in the seasonal cycle (Trenberth et al. 1998). In contrast, the tropical Atlantic warm SSTs have been shown to produce a Gill-response over Central America, similar to Fig. 6c (Fig. 8 of Wang et al. 2008).

Trenberth, K. E., Branstator, G. W., Karoly, D., Kumar, A., Lau, N. C., & Ropelewski, C. (1998). Progress during TOGA in understanding and modeling global teleconnections associated with tropical sea surface temperatures. *Journal of Geophysical Research: Oceans*, 103(C7), 14291-14324.

Lee, S. K., Lopez, H., Chung, E. S., DiNezio, P., Yeh, S. W., & Wittenberg, A. T. (2018). On the fragile relationship between El Niño and California rainfall. *Geophysical Research Letters*, 45(2), 907-915.

Lopez, H., & Kirtman, B. P. (2019). ENSO influence over the Pacific North American sector: Uncertainty due to atmospheric internal variability. *Climate dynamics*, 52(9), 6149-6172

Wang, C., Lee, S. K., & Enfield, D. B. (2008). Climate response to anomalously large and small Atlantic warm pools during the summer. *Journal of Climate*, 21(11), 2437-2450.

- ll. 318-319: *the role of topography seems to be crucial here. Can the authors comment on this?*

Response: Topography plays a major role in the generation of Rossby waves via vortex stretching (Sardeshmukh and Hoskins 1988). We computed the Rossby wave source (RWS, see equation 6 in the revised text Method section), streamfunction and velocity potential to show a more detailed description of the dynamics involved.

The RWS is computed for each SST sensitivity experiment. For example, note that there are a few regions of anticyclonic RWS in the GBL23 experiment (Fig. R13a) over the equatorial Pacific and Atlantic, near the location of the ITCZ, and also over the Greater Antilles. These are consistent with the warm 2023 SST forcing prescribed in the GBL23 experiment, which led to upper level 200 hPa divergence (Fig. R13c) and 200 hPa anticyclonic circulation consistent with a Gill-type response (Fig. R13b). Further decomposition of the RWS into its component demonstrates that mean vortex stretching by the

anomalous divergent flow (Fig. R13e) was the main driver of the anticyclonic RWS and thus the anticyclonic circulation over Mexico.

As the reviewer suggested, topographic features induced cyclonic RWS over western Mexico and southwestern U.S. (Fig. R13), mostly driven by mean vorticity advection and mean vortex stretching source terms (see Methods for decomposition of the RWS).

In addition, this region is home to fairly tall topographic features, like the Sierra Madre, which steers low-level moisture modulating the North American Monsoon (Duan et al. 2024). Topography is also crucial in this region where convection typically develops in the boreal summer over high-elevation terrains (King and Balling 1994) and has been shown to play a major role in steering low-level flow, which feeds the monsoon (Collier and Zhang 2007; Pascale et al. 2016)

We added this discussion and the analysis on the RWS (Lines 371 - 389) and added Fig. R13 to the revised manuscript as Fig. 8. We also added Figs. R14 and R15 to the supplementary materials (supplementary figs. 10 and 11).

Collier, J. C., & Zhang, G. J. (2007). Effects of increased horizontal resolution on simulation of the North American monsoon in the NCAR CAM3: An evaluation based on surface, satellite, and reanalysis data. *Journal of climate*, 20(9), 1843-1861.

Duan, S., Ullrich, P., & Boos, W. R. (2024). Meteorological drivers of North American monsoon extreme precipitation events. *Journal of Geophysical Research: Atmospheres*, 129(13), e2023JD040535.

King, T. S., & Balling Jr, R. C. (1994). Diurnal variations in Arizona monsoon lightning data. *Monthly Weather Review*, 122(7), 1659-1664.

Pascale, S., Bordoni, S., Kapnick, S. B., Vecchi, G. A., Jia, L., Delworth, T. L., ... & Anderson, W. (2016). The impact of horizontal resolution on North American monsoon Gulf of California moisture surges in a suite of coupled global climate models. *Journal of Climate*, 29(21), 7911-7936.

Sardeshmukh, P. D., & Hoskins, B. J. (1988). The generation of global rotational flow by steady idealized tropical divergence. *Journal of Atmospheric Sciences*, 45(7), 1228-1251.

- ll. 348-350: I find this argument a bit illogic. I see no reason why the sum of percentages from ATL and PAC sensitivity experiments would result in percentage increase of GLB23 minus the residual ("synergic interbasin interactions"). It might be informative to obtain composite mean maps for the three experiments, as in Figure 7, then subtract from the complete SST prescribed field to have an idea of the role of interbasin interactions. I do not think though, that the same can be done to infer the relative role of the three components for the overall GLB23 increase in extremes. The role of Atlantic and Pacific alone could determine a growth in the occurrence of the chosen events that would be larger than the combined effects, i.e. the interbasin interaction role would be effectively to reduce the role of the individual basins. This would not be captured if using the described approach, and it would be particularly relevant when dealing with extreme events. Please clarify this;

Response: This is a very important comment that needs further clarification and a more detailed description of the approach which was not adequately provided in the original text. Therefore, we address each concern separately in the discussion below, which has been added to the Supplementary Information of the revised paper.

I see no reason why the sum of percentages from ATL and PAC sensitivity experiments would result in percentage increase of GBL23 minus the residual (“synergic interbasin interactions”)

Response: We first explain how equation 6 of the paper was derived. Using the AGCM sea surface temperature anomaly (SSTA) sensitivity experiments, any variable (e.g., heat wave days, number, surface temperature) response to the prescribed forcing can be extracted as follows:

The climatological occurrence of any event (heat wave number, days) is first extracted from the control (CTL) simulation (equation R1). Note that we obtained this climatology from the 300 year-long CTL simulation so as to get an accurate expected value in the absence of any SSTA forcing. Then, the occurrences of such events in the SSTA sensitivity experiments are thus governed by eq. R2 for the Atlantic and eq. R3 for the Pacific experiments. For the case of both Atlantic and Pacific SSTA forcing (i.e., the GBL23 experiment), a linear combination of ATL23 and PAC23 is not sufficient as pointed out in the paper and by the reviewer, given that there could be interbasin interactions between the ATL and PAC SST forcing, i.e., the Synergy term in eq. R4.

$$CTL = \text{Climatological expected occurrences} \quad (\text{eq. R1})$$

$$ATL23 = CTL + \text{Atlantic SSTA sensitivity} \quad (\text{eq. R2})$$

$$PAC23 = CTL + \text{Pacific SSTA sensitivity} \quad (\text{eq. R3})$$

$$GBL23 = CTL + \text{Atlantic SSTA} + \text{Pacific SSTA sensitivities} + \text{Synergy} \quad (\text{eq. R4})$$

Subtracting eq. R2 and R3 from R4, we obtain the interbasin synergy component:

$$\text{Interbasin Synergy} = GBL23 - ATL23 - PAC23 + CTL \quad (\text{eq. R5})$$

Thus, the sum from ATL and PAC sensitivity experiments would result in the GBL23 minus the residual (i.e., synergic interbasin interactions) plus a climatology, which is present in all experiments.

The percentages shown in Fig. 10 (of revised text) are relative to the CTL simulation i.e., an increase/decrease relative to the expected climatology. For each of the sensitivity experiments (ATL23, PAC23, and GBL23), these could be larger or smaller than the CTL. In our case they are all larger for the study region, indicating that SSTAs were responsible for an increase in heat waves relative to climatology. This increase is quantified as follows:

As an example, for the heat wave number (Fig. 10a): There were 77 events in the CTL case and 184 events in the GBL23 experiment, for a total increase of 138%. The entire azimuthal range (angular distance of 360 degrees) of the donut plot represents the GBL23 (e.g., CTL + ATL23 + PAC23 + Synergy) with the individual components scaled azimuthally by their respective fractions. These were computed from the ATL23 experiment which produced 135 heat waves and from the PAC23 which produced 100 heat wave events, an increase of 75% and 31% from the climatology of 77 heat wave

events respectively. This leaves 32% unaccounted for in the 138% increase seen in the GBL23 case, thus it is attributed to the synergy component.

It is worth noting that all experiments (GBL23, ATL23, and PAC23) contain a climatological component which is the same as the CTL simulation (i.e., same model physics, biases, etc.). These values are the sum (composite) from multi-ensemble simulations (100 ensembles in total with integration length of 1 year) mimicking 100 years for each experiment case. Thus, 77 (184) heat wave events consist of less than one (almost two) events per year in the CTL (GBL23) case.

The approach used here to extract changes in the occurrence of an event is widely used in the attribution analysis (Pearl 2000; Hannart and Naveau 2018; Lopez et al. 2024). For example, when employing causal counterfactual theory to assess the probability that an event E would not have occurred in the absence of a cause I , where event E is defined as a heat wave and C is defined as a warm Atlantic and/or Pacific SST anomalies. Probability of necessary causation (PN) can thus be constructed to measure the fraction of extreme events attributed to C , defined as (eq. R6):

$$PN(C \rightarrow E) = \max \left\{ 1 - \frac{P_0}{P_1}, 0 \right\} \quad (\text{eq. R6})$$

where P_0 is the probability of an event occurring in the counterfactual world (e.g., our control simulation CTL), and P_1 is the probability of that same event occurring in the factual world (e.g., GBL23, ATL23, PAC23 sensitivity experiments). For the case of heat wave number (Fig. 10a), $P_0 = 77$ per 100 years and $P_1 = 184$ per 100 years.

The relation in eq. R6 is analogous to the relative change or percentages shown in eq. R7:

$$\text{Relative change} = (\text{Experiment} - \text{CTL})/\text{CTL} \quad (\text{eq. R7})$$

We added the above explanation of the synergy component along with equations R1, R2, R3, R4, and R5 to the Supplementary Information of the revised paper.

The role of Atlantic and Pacific alone could determine a growth in the occurrence of the chosen events that would be larger than the combined effects, i.e. the interbasin interaction role would be effectively to reduce the role of the individual basins

Agree. The role of the combined Atlantic and Pacific alone (i.e., ATL23 and PAC23) could be larger than the GBL23, thus suggesting that the synergy interaction would have to be negative (see equations R4 and R6). Note that we do not make any assumptions when extracting the synergy effect. In fact, this effect is non-existent with regards to the duration of the heat waves (Fig. 10c) but positive for all other indices e.g., heat wave number (Fig. 10a), heat wave days (Fig. 10b), and frequency (Fig. 10d). With that said, we acknowledge that Fig. 10 should not be generalized for other regions where CTL could produce more events than the GBL23 experiments (i.e., less heat waves due to the SST sensitivity). However, Fig. 10 is a concise representation of the fractional increase in heat wave characteristics due to specific SST anomalies prescribed in the AGCM experiment.

Pearl, J. (2000). Models, reasoning and inference. *Cambridge, UK: Cambridge University Press*, 19(2), 3.

Hannart, A., & Naveau, P. (2018). Probabilities of causation of climate changes. *Journal of Climate*, 31(14), 5507-5524.

Lopez, H., Lee, S. K., West, R., Kim, D., Foltz, G. R., Alaka Jr, G. J., & Murakami, H. (2024). Projected increase in the frequency of extremely active Atlantic hurricane seasons. *Science Advances*, 10(46), eadq7856.

It might be informative to obtain composite mean maps for the three experiments, as in Figure 7, then subtract from the complete SST prescribed field to have an idea of the role of interbasin interactions.

Please note that this is already done by the composite maps shown in Fig. 7g,h. The role of the interbasin synergy interaction is to exacerbate the warm surface temperatures over northern Mexico and the southwestern U.S. This warmer temperature is surrounded by cooler temperatures over the Pacific coast and the eastern U.S. (Fig. 7h).

- I. 375: once again, it is not clear what you mean by “western North America”, and I do not think that your arguments are equally valid for the whole of western North America, e.g. including Canada and Alaska;

Response: We use the term “western North America” loosely in the paper, which we agree could bring some confusion as we are definitely not including Canada and Alaska here. We now clarified in the revised text that the focus is on the 2023 heatwave that impacted the southwestern U.S. and northern Mexico. Any geographical index (e.g., area averaged variable) is explicitly defined in the revised text method section and figure captions.

- II. 412-419: this paragraph does not seem to belong to conclusions, as it would be better fitting into Introduction;

Response: We agree with the reviewer that this paragraph reads more like introductory material. However, since the focus of the paper and the special call targets extremes during 2023, we felt that mentioning extreme conditions that occurred in 2024 would steer the reader away from the paper’s focus. With that said, we remove the paragraph to make the paper more focused on 2023.

- II. 426-427: is there a specific reason why authors decided to use this Reanalysis product instead of others? What is the spatial resolution of this product?

Response: As we mentioned in an earlier comment, the motivation for using the JRA55 analysis is that it provides various diabatic heating rate fields at pressure levels as diagnostic variables, which other reanalysis products do not offer. These heating rates are used to diagnose the thermodynamics of the heat wave event.

The JRA55 spans the period from 1958 to present with a 55 km spatial resolution, and 60 vertical levels up to 0.1 hPa. Data interpolated to $1.25^\circ \times 1.25^\circ$ spatial resolution and 37 vertical levels from 1,000 hPa to 1 hPa are available to download. We clarified this in the Method section (Lines 492 - 500).

- ll. 451-452: the choice of 850hPa temperature seems a bit arbitrary, 2-metres temperature would have been a more valid alternative for near-surface temperature;

Response: Please note that we chose to show the temperature anomaly at the 850hPa level to be consistent with the pressure level at which the heat budget analysis is performed. Quantifying the heat budget near the surface (2-meters) would result in too much emphasis on the surface net heat fluxes, obscuring the role of advective and diffusive heating terms.

With that said, we agree that the 2-meter air temperatures are more relevant for characterizing impacts of extreme heat. Thus, we have modified Fig. 2 to include both the 2-meter and the 850hPa temperatures next to each other. Note that their anomalies are fairly similar partly due to the dominance of vertical advective and diffusive heating and a lack of horizontal temperature advection, features common under stagnant weather i.e., heat dome conditions. The 850hPa temperature anomalies persist a bit longer into August 2023 probably due to longwave heating at 850hPa and reduced surface shortwave heating (Fig. 2).

- ll. 472-473: although I can understand that for the sake of comparison with Reanalyses, it was necessary to choose a reference period that was also encompassed by JRA55, it is a bit odd that a climatology for a period characterized by a strong positive trend and so close to the 2023 event has been considered. Perhaps a previous period (same length, still encompassing the 1958-present JRA55 coverage) would have been a more appropriate choice;

Response: The SST sensitivity experiment is designed this way to explicitly isolate the exceptional SST anomalies of 2023 relative to the recent climatological period (i.e., 1979 – 2022). Note that the goal is for evaluating interannual variability changes in SSTs and its influence on atmospheric circulation and heatwaves. Extending the climatological period back to 1958 would complicate the interpretation of the model runs when comparing 2023 SST-forced runs versus the climatology run. This is because the observed long-term trend will also need to be considered as well as the roles of multidecadal climate variations (e.g., the Pacific decadal oscillation, Atlantic multidecadal variability) along with external forcing from natural and anthropogenic causes.

With that said, we repeated the model experiment (GBL23) but prescribed the 2023 SST anomalies relative to an extended climatology period (1958 – 2022). We compared this extended climatology run to the original GBL23 run (Fig. R9). Besides some small changes, the overall atmospheric response is very similar between the two SST sensitivity experiments. Some minor changes are expected as the longer climatological period further exacerbates the model response to the 2023 SST forcing i.e., from interbasin warming (warmer SST anomalies), due to the global SSTs experiencing a positive trend. For example, the extended climatology experiment produced more heat wave events, days, and larger amplitude than the GBL23 experiment (Table R1).

Another motivation for using the more recent period for climatology is due to the quality of observational records before the satellite era, especially for remote regions of the world oceans. The effect of the longer SST trends is excluded from the revised text; however, we are investigating this in a separate study.

Table R1. Similar to Supplementary Table 1 and showing the changes in heat wave characteristics from the CTL (climatology), GLB23 (2023 SST anomaly relative to 1979 – 2022 period), and GBL23-Extended climatology (SST anomalies relative to the 1958 – 2022 period) AGCM experiments for JJA maximum for the grid-point closest to Phoenix, Arizona.

Experiment	Heat wave Characteristics			
	Number	Days	Duration (longest)	Amplitude (strongest)
CTL	77	307	3.9 (9) days	5.1 (7.7) °C
GBL23	184	853	4.6 (14) days	5.3 (8.3) °C
GBL23-Extended Climo	219	1074	4.9 (15) days	5.5 (10.4) °C

Figure R9. SST sensitivity from an atmospheric model experiment. Composite difference of simulated a) 200 hPa temperature (color) and 200 hPa streamfunction (black contour, $10^6 s^{-1}$) and b) 2-meter air temperature from the AGCM experiment with prescribed 2023 global SSTs (GLB23). c) and d) are the same as a) and b) but for the AGCM experiments with 2023 SST forcing relative to an extended climatology (1958 – 2022). The composite differences are with respect to the control experiment (CTL) for JJA.

- I. 477: how is this ensemble created? What initial conditions are chosen, and how distant from each other? This should be made explicit;

Response: Agree. The ensemble runs are created by first integrating the model without any SST anomaly forcing, just prescribing global climatology for the 1979 – 2022 period; this is referred to here as the control (CTL) run with an integration length of 300 years. The last 100 years of the CTL are used as

initial conditions for the ensembles for the GBL23, ATL, and PAC experiments. That is, the experiments are integrated for one year starting on every 1 January from the CTL (e.g., branch simulation), thus creating ensembles with initial conditions that are one year apart from each other. This approach guarantees more than sufficient separation among ensembles as it pertains to weather noise (i.e., the atmospheric initial state is completely different and independent among ensembles) while each ensemble is being forced by the same underlying SST anomalies (i.e., GBL23, PAC, ATL). We have clarified this in the revised text's Method section (Lines 568 - 574).

- ll. 480-481: same as in my previous comment. There are several bootstrapping techniques that could be used for the scope. For the sake of reproducibility, the technique shall be described here;

Response: For statistical significance, we used a Monte Carlo bootstrapping method to determine confidence intervals by subsampling the dataset. All analyses presented are obtained by randomly selecting r samples out of n observations with replacement (eq. R8). This is done 500 times in order to build a significant distribution of composites and assign 95th percentile confidence levels. This is added in the Method section of the revised manuscript (Lines 601 - 606).

$$\binom{n}{r} = \frac{n!}{r!(n-r)!} = \text{possible combinations} \quad (\text{eq. R8})$$

- ll. 483-484: it is not clear if ATL23 and PAC23 are ensembles as well;

Response: All model experiments were performed as ensemble simulations. We have clarified this in the revised manuscript Method section ((Lines 568 - 574).

References

- Hebert et al. 2022: <https://www.nature.com/articles/s41561-022-01056-4>
- Jiang et al. 2022: <https://agupubs.onlinelibrary.wiley.com/doi/10.1029/2023GL103777>
- Kuhlbrodt et al. 2023: <https://journals.ametsoc.org/view/journals/bams/105/3/BAMS-D-23-0209.1.xml>
- Rex et al. 1950: <https://www.tandfonline.com/doi/abs/10.3402/tellusa.v2i4.8603>
- Richter et al. 2023: <https://agupubs.onlinelibrary.wiley.com/doi/full/10.1029/2024GL111563>
- Stoll et al. 2023: <https://wcd.copernicus.org/articles/4/1/2023/>

These references were added to the revised text.

Reviewer #2 (Remarks to the Author):

Summary:

Overall, this is an interesting and well-written paper. Lopez et al. investigate the extreme and persistent heatwave that impacted the southwestern U.S. and northern Mexico in the summer of 2023. Using observational data, regression analysis, and atmospheric model simulations, the study finds that warm Atlantic SSTs, combined with a developing El Niño in the Pacific, significantly contributed to the increase in heatwave frequency, duration, and intensity. The authors employ heat budget analysis to identify key thermodynamic and radiative processes that sustained extreme surface temperatures. The study concludes that the interbasin synergy between the Atlantic and Pacific basins exacerbated the conditions that led to the record-breaking heatwave. The findings suggest that continued ocean warming may lead to an increased likelihood of prolonged and extreme heat events in the future. I do not have any substantial comments about the analysis or the main conclusions, which seem solid and well-supported. I recommend some minor revisions before this manuscript can be published, please see my comments below.

Response: We are very grateful for your constructive comments/suggestions and hope to have successfully addressed all your comments. Please find our replies to your specific comments below in regular fonts and your original comments in blue italic fonts. We point to specific line numbers where the changes were made, these line numbers refer to the lines on the tracked-changes version of the text

Main comments:

1. Some of the narrative and factual statements in the abstract and introduction can use finetuning, and I have left a number of specific line comments with my suggestions for changes.

Response: We have implemented the suggested changes in the revised manuscript.

2. The multitude of different time periods used for baselines seems unnecessary. For example, depending on the variable, the authors use 1979-2022 (SST and vertically integrated moisture transport), 1982-2023 (Phoenix temperatures), 1979-2023 (heat budget analysis and NAM), and 1960-2023 (SST regression). Could the authors find one or maybe at most two common baselines to use for all the analyses?

Response: Agree. We have now synthesized all station data baseline periods for 1955 - 2023 to match the Japanese 55-year reanalysis data. Note that the 1982-2023 Phoenix temperatures were a typo in the original figure 1 caption, the histogram represents the 1955 - 2023 period. This is fixed in the revised text.

For the reanalysis data (SSTs, 3-dimensional atmospheric variables, and monsoon analysis), we now use the 1979 - 2022 baseline period to define climatology.

Also note that for the SST sensitivity atmospheric model experiments, we intentionally used the 1979 - 2022 period for the climatological run so that to avoid aliasing the prescribed sensitivity runs based on the observed 2023 SSTs.

We clarified this in the Method section of the revised manuscript (Lines 504 - 505).

Specific comments:

Line 14: WMO 2023 reference not in citations.

Response: This is in reference to the State of the Climate in 2023, which is listed as reference 1 and now updated to the State of the Climate in 2024. We added the appropriate reference to the abstract (Line 14).

Blunden, J., & Boyer, T. (2024). State of the Climate in 2023. *Bulletin of the American Meteorological Society*, 105(8), S1-S484.

Line 18: Hazard >> disaster

Response: Changed.

Line 18: Was this the costliest natural disaster of 2023 in North America? Perhaps specify this. Globally, I believe the Turkey earthquake in February was more costly.

Response: Agree, this is confined to just North America. We specify this in the revised text (Line 18).

Line 32-33: Is there a citation for “200 deaths”? I believe the death toll in Maricopa County alone was more than 200.

Response: We have updated the death toll number based on more current studies and government official reports. Below is the modification to the revised manuscript (Lines 33 – 38).

The year 2023 produced the highest number of heat-related deaths in the United States in the 21st century, with 2325 deaths (Howard et al. 2024). One of these severe heat wave events occurred over the southwestern United States (US) and Mexico, which extended from mid-June to early August, affected over 100 million people, and was responsible for 303 deaths that occurred in a span of just two weeks in Maricopa County, Arizona (<https://www.maricopa.gov/1858/Heat-Surveillance>).

Howard, J. T., Androne, N., Alcover, K. C., & Santos-Lozada, A. R. (2024). Trends of heat-related deaths in the US, 1999-2023. *JAMA*, 332(14), 1203-1204.

Line 35: Similar to my earlier comment, it should be specified that this was the costliest natural disaster in North America, not globally.

Response: We specified in the revised text that this figure is only for North America (Line 43).

Line 36-37: I think the more relevant statistic to highlight is the record streak of days above 110F in Phoenix (31 straight), which broke the previous record of 18 straight days. Days >40°C may not be the most relevant statistic for Phoenix where the average high temperature in July is 41.4°C, meaning that 40°C is a cool day. The record warm Tmin of 97°F is interesting and can be kept in the text.

Response: Great point, 40°C is not that impressive of a temperature for the area. We have now changed it to stress the consecutive days of 43.3°C (110°F), which occurred from 30 June - 30 July of 2023 in the revised text (Lines 44– 45).

Line 38-40: This sentence, as currently worded, is inaccurate and should probably be deleted. The heat wave that produced the all-time record in New Orleans was a completely separate synoptic event that occurred on August 27 in a different region of the country. Similarly, the San Juan record resulted from a different tropical heatwave that was not part of the western North American heatwave that the authors are studying.

Response: Agree, the original intention was to stress on the anomalous warm summer of 2023, but these specific events were not related to the heat wave being studied in this paper. We have removed this sentence from the revised text.

Line 60: delete “the”

Response: Done

Line 73: Might be good to add a quick sentence that explains how a Gill-type response can produce blocking over the Great Plains. I am assuming that the broad readership of Nature Communications will not be familiar with this.

Response: Agree. We have now added some references on the Gill-response (Lines 83 – 86).

“A recent work found that boreal summer tropical Atlantic SST anomalies modulate heat wave occurrences over North America (Lopez et al. 2022). In that work, it was found that a warmer tropical Atlantic enhances atmospheric convection over the Caribbean Sea and produces a Gill-type atmospheric response (Gill et al. 1980). This in turn produces an anticyclonic Rossby wave source over the Great Plains, thus enhancing subsidence and significant surface warming, leading to heat domes.”

Note also that we added an additional figure and analysis describing the large-scale atmospheric response to prescribed SST forcing via Rossby wave sources (See Fig. 8 of the revised manuscript).

Line 83-84: I would delete this sentence, since the AMO itself has been called into question as a “real” mode of internal climate variability.

Response: We agree, the current literature consensus calls for reframing the formerly known AMO into Atlantic Multi-decadal variability (AMV) as to not make any assumptions on whether its multi-year

variations are deemed as an internal oscillation. We have updated the revised text and replaced AMO with AMV (Line 101).

Line 79-93: The effect of the 2020 international shipping sulfate reductions over the N. Atlantic might need to be mentioned in this paragraph, since they likely played a role in the warm SST during 2023.

Response: We added the following to the revised manuscript (Lines 104 - 106).

“Furthermore, the 2020 emission regulations from the International Maritime Organization aimed at reducing ship sulfate aerosol emissions may have contributed to the recent warm surface temperatures over the North Atlantic (Gettelman et al. 2024, Manshausen et al. 2023).”

Gettelman, A., Christensen, M. W., Diamond, M. S., Gryspeerdt, E., Manshausen, P., Stier, P., ... & Yuan, T. (2024). Has reducing ship emissions brought forward global warming? *Geophysical Research Letters*, 51(15), e2024GL109077.

Manshausen, P., Watson-Parris, D., Christensen, M. W., Jalkanen, J. P., & Stier, P. (2023). Rapid saturation of cloud water adjustments to shipping emissions. *Atmospheric Chemistry and Physics*, 23(19), 12545-12555.

Figure 1b: It might be good to adjust the x-axis to make the highest Tmax more visible. For example, Phoenix hit 122F (50C) in 1990 and 121F (49.4C) in 1995 but these temperatures are not visible in the histogram, making it seem like 119F (48.3C) in 2023 was the highest tmax between 1982-2023.

Response: While we agree that it is worth mentioning that two other days saw higher temperatures, representing the 50C that occurred on June 26, 1990 and the 49.4C on July 28, 1995 is very difficult in the histogram given that the x-axis is representing occurrences and these two events are in a bin of their own. We tried to increase the bin number (e.g., refining the intervals to 0.5 and even 0.25C) of the histogram, which is the only way of rescaling the x-axis, but but the histogram becomes discontinuous, appearing like a "broken comb", and does not accurately represent the data

Note that the number of bins for the test and histogram below were chosen based on Sturge's rule by taking into account the size of the data (n = 74 years, 1950 - 2023) and the following relation:

$$N \text{ bins} = 1 + \log_2(\text{number of data points})$$

With that said, we added a sentence clarifying the records of 1990 and 1995 in the revised text (Lines 137 - 138).

Line 113: This is incorrect, the 2023 maximum of 119F/48.3C was tied for the 3rd warmest maximum in Phoenix after 122F in 1990 and 121F in 1995 (see my comment above).

Response: Good point. We have corrected this mistake in the revised manuscript (Lines 137 - 138).

Line 117: "engulfed" >> "affected"

Response: Done

Line 124-125: Why highlight this particular Tmax on Aug 23 in New Orleans? They were even hotter on Aug 27th, when their all-time record high of 105F was reached. Perhaps this should be highlighted instead.

Response: Agree, this is corrected in the revised text (Line 152).

Figure 4a: It is surprising that 2020 doesn't appear as a very strong negative anomaly. This was a well-known failed monsoon season in the SW US, e.g.: Hoell et al., 2022: Record low North American monsoon rainfall in 2020 reignites drought over the American southwest. The authors may want to double check their monsoon index calculations, and if found to be correct, perhaps provide the appropriate caveats as to why their index does not capture certain anomalous years.

Response: Our definition of the North American Monsoon (NAM) follows the concept of the global monsoon (Wang and Ding 2008; Lee and Wang 2014). This index is defined as where the local summer (May - September) precipitation minus the winter (November–March) precipitation exceeds 2.5 mm per day and the local summer precipitation exceeds 55% of total annual precipitation (Lee and Wang 2014). In contrast, studies like Hoell et al., 2022 defined the NAM as an area over northwestern Mexico extending into the southwestern United States, a region known as an extension of the NAM (Adams and Comrie 1997; Pascale et al. 2017, 2019).

This definition of the NAM represents a more coherent and geographically constrained approach (Wang and Ding 2008; Lee and Wang 2014; Wang et al. 2020). For example, Fig. R11 and R12 below shows the precipitation anomaly for the wet season of the NAM during 2020 and 2023 from two datasets. The red stipples indicate the regions that meet the NAM criteria in our study. Note that 2020 was very dry mostly over the southwestern U.S (i.e., the northern extension of the NAM) and away from the core monsoon region (red stipples). This is in contrast with the year 2023 where the core region of the NAM was much drier, although concentrated mostly over Mexico.

Since our NAM index is the area-averaged rainfall over the red-stippled region (as defined in Wang and Ding 2008), our NAM index and that of Hoell et al. 2022 are different, thus showing much drier conditions for 2023 relative to 2020 in our case.

We also added the red stipples to Fig. 4 of the revised manuscript for identification of the monsoon region and also added more details on the monsoon definition in the Methods section (Lines 546 - 550).

Lee, J. Y., & Wang, B. (2014). Future change of global monsoon in the CMIP5. *Climate Dynamics*, 42, 101-119.

Wang, B., & Ding, Q. (2008). Global monsoon: Dominant mode of annual variation in the tropics. *Dynamics of Atmospheres and Oceans*, 44(3-4), 165-183.

Wang, B., Biasutti, M., Byrne, M. P., Castro, C., Chang, C. P., Cook, K., ... & Zhou, T. (2021). Monsoons climate change assessment. *Bulletin of the American Meteorological Society*, 102(1), E1-E19.

Figure R11. June - September (JJAS) precipitation anomaly for a) 2020 and b) 2023. The red stipples indicate the North American monsoon region defined as where the local summer (May - September) precipitation minus the winter (November–March) precipitation exceeds 2.5 mm per day and the local summer precipitation exceeds 55% of total annual precipitation. Dataset from the JRA55 reanalysis.

Figure R12. Same as Fig. R11 but using the ERA5 reanalysis.

Line 197-201: Related to the above comment, the methodological details of the monsoon index calculations should be fully explained in the methods section.

Response: We added more detailed explanations of the monsoon calculations as described in the replies to the above comments in the Method section of the paper (Lines 546 - 550).

Line 246: Please check the “(negative)”, it doesn’t seem to match the rest of the sentence.

Response: Thanks for catching this grammatical error, we have deleted the word “negative” in parenthesis.

Line 261: I am confused by “partial regression”. Do the authors mean partial least squares regression?

Response: Correct, the analysis looks at partial least squares regression to linearly extract independent relationships. This is clarified in the revised text (Line 304).

Line 346: I may have missed it, but I don’t recall seeing the extreme value analysis described in the methods?

Response: Extreme value analysis refers here as the analysis on several heat wave metrics (i.e., occurrence, duration, amplitude) where the heat waves are defined by the excess heat factor (see Method section and equations 1, 2, and 3). However, we agree that the term extreme value analysis is misleading as it is often referred to as a specific branch of statistics dealing with the tail of distributions (e.g., extreme value theory, Coles 1999).

Thus, we have modified the text to reflect this change (Line 412).

Coles, S. (1999). Extreme value theory and applications. Notes from a Course on EVT and Applications Presented at the 44th Reunio Annual da RBRAS e 8th SEAGRO.

Line 387: “persistent heat wave” >> “persistence of the heat wave”

Response: Changed.

Line 416-419: The statement about the earliest 110°F on record in June 2024 is only true for Las Vegas, not Phoenix.

Response: We deleted this text as it relates to anomalous 2024 conditions, which may have distracted the readers from the main scope of the study.

Line 429: “Climate” >> “Centers”

Response: Fixed

Line 501: Delete “the”

Response: Done

Reviewer #3 (Remarks to the Author):

"The longest-lasting 2023 western North American heat wave was fueled by the record-warm Atlantic Ocean" by Lopez et al.

Summary:

This study investigates the causes of the longest-lasting heat wave in western North America in 2023 through observational analysis and atmospheric general circulation model (GCM) experiments. The key conclusion is that the 2023 heatwave was significantly driven by record-warm Atlantic sea surface temperature (SST) and growing El Niño in the Pacific. The authors conducted a comprehensive analysis of the roles of both the atmosphere and the ocean in driving the heat wave, using energy budget analysis, 3-dimensional atmospheric circulation and moisture transport, and the impact of SST over both the Atlantic and Pacific Oceans. I believe this manuscript is well-written and makes a valuable scientific contribution to understanding the dynamic and thermodynamic changes that triggered this heatwave event. However, I found several issues that need to be addressed to improve the manuscript. For publication, please revise the manuscript to satisfactorily address the specific comments provided below.

Response: We are very grateful for your constructive comments/suggestions and hope to have successfully addressed all your comments. Please find our replies to your specific comments below in regular fonts and your original comments in blue italic fonts. We point to specific line numbers where the changes were made, these line numbers refer to the lines on the tracked-changes version of the text

Specific Comments:

1. The authors concluded that the exceptionally warm Atlantic Ocean and the growing El Niño in the Pacific caused changes in circulation, leading to an anticyclonic blocking pattern during the summer of 2023. They supported their findings with evidence of interbasin synergy effect captured in model experiments. However, it remains unclear about how the Pacific SST interacts with the Atlantic in this synergy effect to create the persistent blocking pattern. The dynamic mechanism needs to be addressed as to how the El Niño-driven SST in the Pacific is linked to the ongoing atmospheric blocking over the western U.S.

Response: We made further analysis on the mechanisms involved in the relationship between the Pacific-Atlantic interbasin SST forcing and associated atmospheric response, which led to the blocking pattern over the heat wave region. Since both the Atlantic and Pacific SST forcing are tropical heating anomalies, they could force remote teleconnections. These teleconnections are analyzed via Rossby wave sources described in equation R9 (Sardeshmukh and Hoskins (1988) and Kirtman et al. (2001)), where the anomalous RWS is defined by the anomalous vorticity advection by the mean divergent wind, mean vorticity advection by the anomalous divergent winds, anomalous vortex stretching, and mean vortex

stretching, respectively. Here, u_i is the divergent wind (i.e., irrotational component of the flow), ζ is the relative vorticity, and f is the Coriolis parameter. Overbar denotes the mean climatology and primes represent the deviation from climatology.

$$RWS = -\nabla(u_i \zeta)' = -\bar{u}_i \cdot \nabla \zeta' - u' \cdot \nabla(\bar{\zeta} + f) - \zeta' \nabla \cdot \bar{u}_i - (\bar{\zeta} + f) \nabla \cdot u_i' \quad (eq. R9)$$

The RWS is computed for each SST sensitivity experiment. For example, note that there are a few regions of anticyclonic RWS in the GBL23 experiment (Fig. R13a) over the equatorial Pacific and Atlantic, near the location of the ITCZ, and also over the Greater Antilles. These are consistent with the warm 2023 SST forcing prescribed in the GBL23 experiment, which led to upper level 200 hPa divergence (Fig. R13c) and 200 hPa anticyclonic circulation consistent with a Gill-type response (Fig. R13b). Further decomposition of the RWS into its component demonstrates that mean vortex stretching by the anomalous divergent flow (Fig. R13e) was the main driver of the anticyclonic RWS and thus the anticyclonic circulation over Mexico.

The individual roles of each ocean basin are investigated using the ATL23 and PAC23 sensitivity experiments. For the Atlantic-only forcing (Fig. R14), a similar pattern emerges where the upper-level divergent flow being responsible for the anticyclonic RWS over the Atlantic basin, which leads to a downstream (westward) intensification of the upper-level anticyclone over Mexico and the eastern Pacific between 20 - 40°N. In contrast, the Pacific-only SST sensitivity experiment (Fig. R15) yields anticyclonic RWS over the ITCZ, in association with the developing El Niño, and consistent with upper-level divergence over the heating region (Trenberth et al. 1998). However, this signal is smaller than those from the ATL23 forcing. The larger RWS from the ATL23 compared to the PAC23 sensitivity experiment is probably due to the fact that SSTs in the Atlantic were at their peak while the El Niño in the Pacific was still in the growing phase (Supplementary Fig. 3).

We added this discussion to the revised text (Lines 371 - 389) and figure R13 as Fig. 8 of the revised text. We also added figures R14 and 15 to the supplementary Figs. 10 and 11. The RWS analysis and equation R9 were added to the Method section (Lines 762 – 769).

- Kirtman, B. P., Paolino, D. A., Kinter III, J. L., & Straus, D. M. (2001). Impact of tropical subseasonal SST variability on seasonal mean climate simulations. *Monthly weather review*, 129(4), 853-868.
- Sardeshmukh, P. D., & Hoskins, B. J. (1988). The generation of global rotational flow by steady idealized tropical divergence. *Journal of Atmospheric Sciences*, 45(7), 1228-1251.
- Trenberth, K. E., Branstator, G. W., Karoly, D., Kumar, A., Lau, N. C., & Ropelewski, C. (1998). Progress during TOGA in understanding and modeling global teleconnections associated with tropical sea surface temperatures. *Journal of Geophysical Research: Oceans*, 103(C7), 14291-14324.

Figure R13. SST sensitivity from an atmospheric model experiment for the Rossby wave source (RWS) components and circulation. Composite difference of simulated a) 200 hPa RWS, b) 200 hPa streamfunction (contour, $10^6 s^{-1}$) and anomalous vorticity advection (color), c) 200hPa velocity potential (contour, $10^6 s^{-1}$) and mean vorticity advection (color), d) 200 hPa rotational wind (vector, ms^{-1}) and anomalous vortex stretching (color), and d) 200hPa divergent wind component (vector, ms^{-1}) and mean vortex stretching. The units for the RWS terms are $10^{-11} s^{-2}$, see Methods for definition. Composites are from the AGCM experiment with prescribed 2023 global SSTs (GLB23). The composite differences are with respect to the control experiment (CTL) for June-July-August.

Figure R14. Same as R13 but for the ATL23 model experiment which compares the sensitivity to Atlantic SST forcing.

Figure R15. Same as R13 but for the PAC23 model experiment which compares the sensitivity to Pacific SST forcing.

2. *Potential driver of the Atlantic warming in the discussion section: I'm still not convinced about the possible impact of El Niño on the Atlantic SST increase (e.g., L404-406), as warmer-than-average Atlantic SST was not frequently observed during the previous El Niño developing years. Could you provide a more compelling explanation of the El Niño-driven increase in Atlantic SST?*

Response: Please note that we are not arguing for an El Niño event being the cause of the exceptional warm Atlantic SSTs, just pointing out that it could be one of the contributors as described in Cheng et al. 2019 and in Li et al. 2024. For example, during El Niño events, the tropical troposphere is generally warmer globally, which modulates surface winds and surface heat fluxes that eventually shoals the tropical ocean mixed layer, making it more stable, which leads to surface warming.

However, we agree that El Niño alone could not cause such extreme north Atlantic SST anomalies observed in 2023 (which were exceptionally warm to the extent of 3 standard deviations above the mean climatology). Other factors that could play more relevant roles are the Atlantic Ocean internal variability (i.e., Atlantic Multi-decadal variability or AMV) and external forcing from natural or anthropogenic effects like aerosols (Booth et al. 2012; Bellucci et al. 2017). For example, recent work found that emission regulations from the International Maritime Organization aimed at reducing ship sulfate aerosol emissions may have contributed to the recent warm surface temperatures over the North Atlantic (Gettelman et al. 2024, Manshausen et al. 2023).

We updated the discussion and added more references on the 2023 warming of the North Atlantic in the revised text (Lines 96 - 113 and 462 - 478).

- Bellucci, A., Mariotti, A., & Gualdi, S. (2017). The role of forcings in the twentieth-century North Atlantic multidecadal variability: The 1940–75 North Atlantic cooling case study. *Journal of Climate*, 30(18), 7317-7337.
- Booth, B. B., Dunstone, N. J., Halloran, P. R., Andrews, T., & Bellouin, N. (2012). Aerosols implicated as a prime driver of twentieth-century North Atlantic climate variability. *Nature*, 484(7393), 228-232.
- Cheng, L., Trenberth, K. E., Fasullo, J. T., Mayer, M., Balmaseda, M., & Zhu, J. (2019). Evolution of ocean heat content related to ENSO. *Journal of Climate*, 32(12), 3529-3556.
- Gettelman, A., Christensen, M. W., Diamond, M. S., Gryspeerd, E., Manshausen, P., Stier, P., ... & Yuan, T. (2024). Has reducing ship emissions brought forward global warming?. *Geophysical Research Letters*, 51(15), e2024GL109077.
- Li, K., Zheng, F., Zhu, J., & Zeng, Q. C. (2024). El Niño and the AMO sparked the astonishingly large margin of warming in the global mean surface temperature in 2023. *Adv. Atmos. Sci.* 41, 1017–1022.
- Manshausen, P., Watson-Parris, D., Christensen, M. W., Jalkanen, J. P., & Stier, P. (2023). Rapid saturation of cloud water adjustments to shipping emissions. *Atmospheric Chemistry and Physics*, 23(19), 12545-12555.

3. *This study shows many results for Phoenix, Arizona (e.g., Figures 1, 3b, c, and 9). Extremely high temperatures were, however, observed in other regions as well, including Texas, northern Mexico, and Louisiana. I recommend that the authors provide evidence showing that these other areas share quite similar characteristics to Phoenix, or at least discuss these regions in greater detail.*

Response: Agree. The spatial coverage of the heat waves was fairly large, which encompassed multiple states and most of northern Mexico. Our choice of showing results from just Phoenix was arbitrary but partially motivated by this population center being one of the largest impacted by the event. With that said, we have now added more discussion on the effects of the 2023 heat wave on other important population centers. For example, fig. R16 below is similar to fig. 1a of the paper and shows the temperature evolution during the summer of 2023 for a) Las Vegas, Nevada, b) Albuquerque, New Mexico, c) El Paso, Texas, and d) San Antonio, Texas. Note that all these cities were impacted by the heat wave (supplementary fig. 1) with both maximum and minimum temperatures exceeding their corresponding 95 percentile for multiple days.

Also note that the analysis only includes cities within the continental U.S. as long-term continuous and reliable data are not available for northern Mexico.

We added Fig. R16 as a supplementary Fig. 1 and associated discussion in the revised text (Lines 140 – 144)

Figure R16. Heat wave event of 2023. Seasonal evolution of maximum (red) and minimum (blue) temperature for the year 2023 from 1 June to 20 September for a) Las Vegas, Nevada, b) Albuquerque, New Mexico, c) El Paso, Texas, and d) San Antonio, Texas. The long-term daily mean is shown by the dashed line whereas the 5th and 95th percentiles are shown by the shading region. Excess above the 95th

percentile is shown by red shading for both maximum and minimum temperatures. The observed histogram of maximum (red) and minimum (blue) temperature for the period 1 June to 31 August from 1955-2023 is also shown on the right with bin size of 1°C. The extremely warm temperatures for the 2023 summer are shown for reference.

4. Equation 4 for budget analysis: I believe vertical integration over a range of pressure levels would be more preferable than considering a single level (850hPa). This would be specifically true for vertical advection.

Response: We were following the approach of Wei et al. 2023 by computing the heating rates at a pressure level relatively low enough to represent near-surface conditions. However, we agree that integrating over a range of pressures levels close to the surface would be preferable, so we have recomputed the energy budget integrated from 975 to 800 hPa (equation R10), which includes the following pressure levels: 975, 950, 925, 900, 875, 850, 825, and 800 hPa. The integral is mass-weighted and normalized by the total integration thickness ΔP to preserve the original units of °C/day of the heating rates and for easier comparison with the original units.

$$\frac{1}{\Delta P} \int_{975}^{800} \frac{\partial T}{\partial t} dp = \frac{1}{\Delta P} \int_{975}^{800} \left[-\vec{u} \cdot \nabla T - \omega \left(\frac{\partial T}{\partial p} - \frac{RT}{C_p P} \right) + Q_{net} \right] dp \quad (eq. R10)$$

Note that the overall results and the relative contributions of each of the heating terms do not change (Fig. R17), and there is only a small increase in the role of the horizontal advection term leading up to the heat wave (June) and later in the period (second half of August).

We have replaced Fig. 2 with Fig. R17 below and updated equation 4 in the revised text (Lines 525 - 533) to represent the vertical integration. Note that we have also added a new sub-panel showing the 2-meter air temperature anomalies as requested by another reviewer.

Wei, J., Wang, B., Luo, J. J., Li, C., & Yuan, C. (2023). Synoptic characteristics of heatwave events in Australia during austral summer of 1950/1951–2019/2020. *International Journal of Climatology*, 43(12), 5662-5680.

Figure R17. Heat budget analysis. Energy budget averaged every 10 days over the southwest U.S. and northwest Mexico from 6 June to 24 August 2023. Each row represents (from the top): 200 hPa geopotential height anomaly [gpm], 850 hPa temperature and 2-meter temperature anomalies [°C], vertically integrated anomalous heating rates from 975 - 800 hPa [°C day⁻¹], and surface heat fluxes (net surface shortwave and longwave radiation, sensible and latent heat fluxes) [Wm⁻²]. Daily anomalies are derived from the long-term monthly mean for the 1979-2022 period.

5. Advection in budget analysis: Could you elaborate which factor, the wind versus temperature gradient, and mean versus anomaly, has a greater influence on determining the advection in the budget analysis?

Response: We decomposed the advection term and found that the anomalous advection by the anomalous wind contributed the most to the total advection.

The relative contribution of wind (temperature gradient) was assessed by first computing the root-mean-square amplitude of the temperature gradient (winds) shown in brackets (e.g., time-invariant vector), then multiplied by the wind (temperature gradient) anomalies shown by primes. Overall, the wind anomalies proved to be more important than the anomalous temperature gradient (Fig. R18).

With that said, note that heating due to advection is very small during the peak of the event (July), this is expected due to the broad temperature anomaly (Fig. 1c) and stagnant flow pattern (Fig. 3) associated with this heat wave.

Please note that we did not include Fig. R18 in the revised text due to the relatively small contribution of the horizontal advection to the heat wave and the already large number of figures presented. However, we mention the relative contributions of the advection terms in the revised text (Lines 180 - 181).

Figure R18. Decomposition of the total anomalous temperature advection averaged every 10 days over the southwest U.S. and northwest Mexico from 6 June to 24 August 2023. Each row represents (from the top): anomalous temperature advection, anomalous temperature advection by the mean wind, mean temperature advection by the anomalous wind, anomalous temperature advection by the anomalous wind, the role of circulation-only, and the role of temperature gradient only in the horizontal advection. All values were vertically integrated from 975 - 800 hPa [$^{\circ}\text{C day}^{-1}$]. Daily anomalies, denoted by primes, are derived from the long-term monthly mean climatology for the 1979-2022 period, denoted by the overbar. The entries within brackets are defined by the root mean square amplitude from June-August, effectively holding their contribution to the advection constant.

6. The authors mention that they present the PV pattern in Figure 3 during the peak of the heatwave. However, it seems this is based on the heatwave that occurred in Arizona. I am wondering how the large-scale atmospheric condition shown in Figure 3 could support the development of heatwave conditions over E Mexico, E Texas and Louisiana, where northerly winds are prevalent. Is the PV pattern here explaining the large-scale condition for the heatwave over Arizona only?

Response: The PV analysis shown in Fig. 3a and b) encompasses the broad scale of the heat wave at its peak. However, this heat wave was not confined to Arizona, we picked this specific longitude cross-section to represent the upper-level dynamics. However, similar large-scale atmospheric conditions were present during the early (Fig. R19) and late stages (Fig. R20) of the heat wave. For example, a deep anticyclonic blocking pattern and warm near-surface potential temperature anomalies were also present. This is consistent with the semi-persistent anticyclone and long-duration of the heat wave (Supplementary Fig. 4).

Figure R19. Similar to figure 3 of the manuscript. a) Potential vorticity and wind at the 340K isentropic level during the maximum amplitude of the heat wave on 20 June 2023. Thick vectors depict anti-cyclonic fluid trapping, a proxy for heat dome and air flow stagnation. The thick black line indicates the location of the dynamical tropopause. The blue star represents the location of San Antonio, Texas. b) Latitude-vertical cross-section along 100°W on 20 June 2023 of anomalous potential temperature (color) and potential temperature (magenta 10K intervals). Also shown are zonal wind anomalies (light black contours at 3 m/s intervals), the tropopause level as measured by the 2 PVU (thick black line), and anti-cyclonic fluid trapping (circle hatching at 10^{-5} s^{-1}).

Figure R20. Same as figure R12 but for 06 August 2023. The cross section is shown along 100°W , near El Paso, Texas.

7. Following the comment above (#6), the timing of the heatwave peak seems to vary across different regions, such as Arizona, Texas, Louisiana, NW Mexico, and E Mexico, as seen in Supplementary Figure 1. The PV and circulation pattern during the peak (June, July, or August) of the heatwave could differ between regions. The authors may need to address this issue.

Response: Please note that the purpose of Fig. 2 is just a descriptive snapshot to gain some insight of the upper-level dynamics involved. We agree that these patterns are time dependent and evolving during the event. As such, and as our reply to your previous comment, we have also analyzed these PV and circulation patterns for different regions and months. These are included for 20 June 2023 (Fig. R19) and 06 August 2023 (Fig. R20). The relevant features for the heat wave (deep anti-cyclonic blocking pattern and warm near-surface potential temperature anomalies) are still present in these different cross-sections at different times.

8. Anomalous moisture transport: Figure caption 4 states this field is computed from departures from climatology. Does this mean that the departure of wind and moisture from climatology is used to compute the anomalous moisture transport and its convergence? If so, I believe this approach may be incorrect. Moisture transport is determined by four components: 1) advection of mean moisture by mean wind, 2) advection of mean moisture by anomalous wind, 3) advection of anomalous moisture by mean wind, and 4) advection of anomalous moisture by anomalous wind. The fourth term may be an order of magnitude smaller than the other terms. I recommend that authors clarify how the Figure 4c pattern was calculated.

Response: The anomalous moisture transport shown in Fig. 4 represents the sum of the last three terms described by the reviewer, which are: advection of mean moisture by anomalous wind, advection of

anomalous moisture by mean wind, and advection of anomalous moisture by anomalous wind. This is clarified in the revised text (Lines 250 - 253). Note that anomalous is referred to here as not including the mean moisture advection by the mean wind, because we are interested in the 2023 anomalous conditions. This is readily shown in equation (eq. R11 and R12) below where moisture flux is the product of the horizontal wind u and specific humidity q and the overbar denotes climatology and primes denotes deviation from climatology:

$$uq = (\bar{u} + u')(\bar{q} + q') = \bar{u}\bar{q} + u'\bar{q} + \bar{u}q' + u'q' \quad eq. R11$$

Thus, what is shown in figure 4 is:

$$\text{Anomalies} = (uq)' = uq - \bar{u}\bar{q} = u'\bar{q} + \bar{u}q' + u'q' \quad eq. R12$$

As the reviewer mentioned, the last term is an order of magnitude smaller. And the moisture transport is dominated by the circulation anomalies ($u'\bar{q}$). This is shown in Figs. R21 and R22 below and clarified in the revised text, which includes a supplementary figure showing the decomposition of the fluxes (Supplementary Figs. 6 and 7).

Figure R21. Decomposition of the anomalous June-August 2023 vertically integrated moisture transport (vector, $Kg m^{-1} s^{-1}$) and its divergence (color, $Kg m^{-2} s^{-1}$), where negative values indicate convergence. a) Advection of mean moisture by anomalous wind, b) advection of anomalous moisture by the mean wind, and c) advection of anomalous moisture by anomalous wind. The overbars denote climatology and primes denote deviation from climatology computed from the departure from the 1979-2022 climatology.

Figure R22. (left-column) Zonal-vertical cross-section along 28°N of the 2023 moisture transport anomaly (color). (Right column) is the same but for the meridional cross-section along 110°W . The transport is decomposed into its components such as (a and b) advection of mean moisture by anomalous wind, (c and d) advection of anomalous moisture by the mean wind, and (e and f) advection of anomalous moisture by anomalous wind. The overbars denote climatology and primes denote deviation from climatology computed from the departure from the 1979-2022 climatology.

9. The moisture transport field suggests that the heat transport from the Atlantic might also be lower than climatology, potentially due to the prevailing wind direction, which is not southerly. Could you offer some discussion on this?

Response: Indeed, the low-level (surface to 600hPa) vertically integrated heat transport anomaly from the Atlantic was reduced in 2023 (Fig. R23). Note that we used equation R12 above for the computation of the anomalous heat transport replacing specific humidity with temperature instead.

Figure R23. a) Climatological vertically integrated moisture transport (vector, $Kg\ m^{-1}\ s^{-1}$) and its divergence (color, $Kg\ m^{-2}\ s^{-1}$), where negative values indicate convergence. b) Same as a) but for the 2023 anomalies computed from the departure from the 1979-2022 climatology. Vertical integration is from the surface to 600hPa.

10. L249: Is it “significantly” negative? The negative anomaly observed in June weakens in July. Additionally, could you explain how this negative anomaly in the Atlantic is related to the anticyclonic circulation linked to atmospheric blocking in the upper troposphere? Are they connected, or is the blocking a consequence of upper-level dynamics with no influence from the surface level shown in Figure 5?

Response: Agree, the sea level pressure anomaly is much weaker in July over the Atlantic as well compared to the June anomalies. We have softened the language here to reflect this weakening.

As per the role of the negative low level pressure anomaly on the upper-level flow, please see our reply to your Comment 1 and the analysis of atmospheric circulation and the Rossby wave source (equation R9). In summary, the warm Atlantic SSTs forced Gill response by creating lower (upper) level convergence (divergence) shown in Fig. R24, which through vortex stretching led to anti-cyclonic Rossby wave source, leading to the atmospheric blocking in the upper troposphere. This mechanism of Atlantic SST forcing is described in Wang et al. 2008.

Wang, C., Lee, S. K., & Enfield, D. B. (2008). Climate response to anomalously large and small Atlantic warm pools during the summer. *Journal of Climate*, 21(11), 2437-2450.

Figure R24. a) 200 hPa streamfunction (black contour, $10^6 s^{-1}$) and velocity potential (color, $10^6 s^{-1}$) anomalies from the AGCM experiment with prescribed 2023 Atlantic SSTs (GLB23). The anomalies are defined with respect to the control experiment (CTL) for June-July-August.

11. The authors present Figure 5 to explore large-scale climate factors that may drive heatwave conditions. However, in terms of causality, it is not clear whether the precipitation over U.S. land is a cause or an effect of the heatwave. This needs to be explicitly addressed in the manuscript.

Response: We added more explanation on the physical processes involved in the persistence of the heat wave in the revised text. For example, the upper troposphere analysis on circulation (rotational and divergent flow, as well as the Rossby wave source RWS analysis) describes how the large-scale SST forcing was linked to the reduced monsoon and heat wave occurrence in the region of interest. See Fig. 8 and associated discussion (Lines 371 - 389).

We also added more analysis showing the historical negative correlation between an inactive monsoon and warm 2-meter temperatures over the southwestern U.S. (discussion in Lines 238 - 241 and Supplementary Fig. 5).

12. Is the upper-level anticyclonic circulation pattern in Figure 6a (contoured) similar to the observed pattern in 2023? If so, is this pattern generally pre-dominant throughout the summer?

Response: Yes, the upper-level circulation pattern from Fig. 6a is very similar to the observed 2023 anomalous pattern as shown in Fig. R25 below. Of course, the anomalous 2023 pattern is a bit noisier as it is just one realization instead of the regression pattern shown in Fig. 6a. Even the surface temperature anomalies are extremely similar (Figs. R25b and d). We added the comparison between the observed 2023 anomalies and those extracted from the regression analysis to the revised Fig. 6.

If so, is this pattern generally pre-dominant throughout the summer?

Please note that these are regression coefficients computed for the June-August period, which depends on the El Niño conditions in the Pacific (Niño3 index) plus the tropical North Atlantic SST anomalies (TNA index). See description in the main text.

We have added panels a and b of Fig. R25 to Fig. 6 of the revised paper as it shows relevant information.

Figure R25. a) Anomalous 200 hPa temperature (color) and streamfunction (black contour, $10^6 s^{-1}$) for June - August of 2023. b) same as a) but for 2-meter maximum air temperature anomaly. c) Regression coefficient of Niño3 plus TNA SSTs and 200 hPa temperature (color) and 200 hPa streamfunction (black contour, $10^6 s^{-1}$). d) Same as c) but for 2-meter maximum air temperatures. The regression coefficients are computed for June-August for the 1960-2023 period through partial regression and the units are per standard deviation of the SST anomalies.

13. Why does the Atlantic-only experiment produce warmer than average temperature along the western coast of Mexico, with no notable warming over the central to eastern side of Mexico (Figure 7)?

Response: Please note that while the larger amplitude temperatures are located on the western portion of Mexico, warm temperatures are found through the country, with the exception of the extreme south, along the Pacific coast (Fig. 7d). In fact, warm temperature anomalies occur all along the U.S. Gulf Coast from Central America to the Florida peninsula, very similar to the GBL23 (all forcing experiment). With that said, we agree that the ATL23 tends to push the anomalies westward compared to the GBL23 case. This is also evident in the circulation anomalies (Fig. 7c) where the upper-level anticyclone is strongest just west of the North American continent (e.g., westerly intensified), compared to the GBL23, which anchors the upper-level cyclone over Mexico. The synergy component appears to be playing an active role in

defining the location of the maximum temperature anomalies, by shifting it inland and north into the Arizona-New Mexico-Texas region (Fig. 7h).

14. While the amplitude change of heatwave events in the AGCM experiments shows the highest values over Mexico, the number of events, heatwave days, and duration in Figure 8 show the greatest increase in the far inland areas like Colorado, as well as western Mexico and the southern coastal region. Have you considered why the largest values appear in the far inland area? Could the large-scale atmospheric pattern in the AGCM experiment offer any insights to help explain this?

Response: This is a great point. The AGCM response to prescribed SST forcing appears more consistent over Mexico, where all four heat wave parameters show consistent increase. In contrast, all parameters except the amplitude are also enhanced over the U.S, consistent with the enhanced duration of the heat wave in 2023.

We examined the large-scale circulation pattern (Figs. R13) and could not deduct a clear reason for such. It is worth mentioning that besides the modulating effects of circulation, heat waves are also influenced by complex land-atmosphere interactions/feedbacks that may not be well resolved in CESM2. In addition, significant topographic features are present in this area, which have been shown to play an important role in surface temperature variations, but may not be well-resolved at these horizontal resolutions. Moreover, complex air-sea interactions are also missing from these AGCM simulations as the atmospheric model is forced by SSTs with no ocean thermodynamic nor dynamical feedback. Thus, further studies may be needed to investigate these differences.

All these caveats and model limitations have been included in the Method section of the revised manuscript (Lines 587 - 593).

Please extend our gratitude to the three anonymous reviewers for their constructive comments/suggestions that have led to a significant improvement to our manuscript.

In what follows, we respond to each comment of all reviewers. Our responses are in regular font and the original reviewers' comments are in blue italic font. Each comment is addressed in the revised manuscript. We point to specific line numbers where the changes were made, these line numbers refer to the lines of the tracked-changes version of the text, A copy of the revised manuscript is provided with tracked changes revisions.

Reviewer #1 (Remarks to the Author):

I acknowledge that the manuscript “The longest-lasting 2023 western North American heat wave was fueled by the record-warm Atlantic Ocean” by Hosmay Lopez, Sang-Ki Lee, Robert West, Dongmin Kim, Liwei Jia featured substantial improvement compared to the previous version. Particularly, I appreciated the great deal of additional work that the authors put into the manuscript, thoroughly addressing comments by myself and the other reviewers (and also proving that I was incorrect about a few remarks!). I do acknowledge that the amount of information provided is now fairly larger and the message is more complete and robust, also involving aspects of the large-scale circulation that were barely touched upon in the previous manuscript. I do have a few minor remaining comments that I want to raise, mainly regarding additional references and better context on a few aspects.

Provided this, I do believe that the manuscript would be ready for publication.

Response: We are very grateful for your constructive comments/suggestions and hope to have successfully addressed all your comments. Please find our replies to your specific comments below in regular fonts and your original comments in blue italic fonts. We point to specific line numbers where the changes were made, these line numbers refer to the lines on the tracked-changes version of the paper.

MINOR COMMENTS

ll. 29-30: given that the authors acknowledge the spatial compounding nature of the event, it would be nice if the authors could mention already here something about the relevance of what they find for the large-scale atmospheric circulation, especially given that they adopt the strategy of looking into inter-basin synergies and the role of planetary-scale Rossby waves (e.g. Lembo et al. 2024).

Lembo, V., and Coauthors, 2024: Dynamics, Statistics, and Predictability of Rossby Waves, Heat Waves, and Spatially Compounding Extreme Events. Bull. Amer. Meteor. Soc., 105, E2283–E2293

Response: We added the fact that several heatwave events were concurrent and affected multiple regions as a part of the large-scale flow characteristics. We also added the reference to Lembo et al. 2024 to the revised text. With that said, mentioning the relevance of our findings this early in the introduction may be a bit premature and steer the focus away from the historical context conveyed in the introduction with regards to how anomalous 2023 was.

ll. 59-61: quasi-resonant amplification is not the only possible explanation for large-scale atmospheric circulation signature on heatwaves (see reference above and literature cited therein);

Response: We added more explanation on large-scale atmospheric circulation signature on heat wave. For example, the role of quasi-stationary and propagating Rossby waves and their interaction with the overall synoptic flow, topography, and land-sea contrast described in Lembo et al. 2024 and references therein. We also added Lembo et al. 2024 to the reference list (ref. 4). See lines 61 – 63 of the revised text.

ll.73-74: I think it is very relevant to mention here a recent paper (England et al. 2025) on the drivers of the North Atlantic SST exceptional anomalies;

England, M.H., Li, Z., Huguenin, M.F. et al. Drivers of the extreme North Atlantic marine heatwave during 2023. Nature (2025). <https://doi.org/10.1038/s41586-025-08903-5>

Response: We added the England et al. 2025 to the discussion section where we discuss drivers of the extremely warm 2023 North Atlantic SSTs, see lines 443 and 446 (ref. 90)

ll. 107-110: In this respect, it is also worth mentioning that the tropical atmospheric bridge here mentioned is linked itself with the extratropical Atlantic variability, as argued for instance in An et al. 2021;

An, X., B. Wu, T. Zhou, and B. Liu, 2021: Atlantic Multidecadal Oscillation Drives Interdecadal Pacific Variability via Tropical Atmospheric Bridge. J. Climate, 34, 5543–5553

Response: We added the An et al. 2021 to the reference list (ref. 59) and revised text, see line 98.

Figure 2: I still think that Figure R2 is actually more informative than the boxes approach (for instance, showing the relative magnitude of anomalies, and some peculiar behavior, such as for instance the one exhibited by latent heat fluxes), but I acknowledge the readability issue and therefore leave it to the authors;

Response: Given the readability issue with the spaghetti/line plot, we choose to keep the box plot version instead.

Reviewer #2 (Remarks to the Author):

I thank the authors for addressing my points. I believe this manuscript is ready to be published pending the approval of the other reviewers.

Response: Thanks a lot for your constructive comments/suggestions, which led to a much-improved manuscript.

Reviewer #3 (Remarks to the Author):

I appreciate the authors for revising their manuscript. My previous concerns have been addressed satisfactorily, and I have no remaining major issues. I recommend accepting the revised version as it stands.

Response: Thanks a lot for your constructive comments/suggestions, which led to a much-improved manuscript.